# Outlier-Robust Phase Retrieval in Nearly-Linear Time

## Abstract

Phase retrieval is a fundamental problem in signal processing, where the goal is to recover a (complex-valued) signal from phaseless intensity measurements. It is known that natural non-convex formulations of phase retrieval do not have spurious local optima. However, the theoretical analyses of such landscape results often rely on strong assumptions, such as the sampling vectors being Gaussian distributed.

In this paper, we propose and study the problem of outlier robust phase retrieval. We seek to recover a vector $x \in \mathbb{R}^d$ from $n$ intensity measurements $y_i = (a_i^\top x)^2$, where the sampling vectors $a_i$'s are initially i.i.d. Gaussian, but a small fraction of the $(a_i, y_i)$ pairs are adversarially corrupted.

Our main result is a near-sample-optimal and nearly-linear-time algorithm that provably recovers the ground-truth $x$ in the presence of adversarial corruptions. We first solve a lightweight convex program to find a vector close to the ground truth. We then run robust gradient descent starting from this initial solution, leveraging recent advances in high-dimensional robust statistics. Our approach is conceptually simple and provides a framework for developing robust algorithms for other tractable non-convex problems.

## 1 Introduction

Phase retrieval is a fundamental problem in signal processing with applications in various fields, including electron microscopy [32], crystallography [33, 36], astronomy [11], and optical imaging [37]. In these applications, one often has access to only the magnitudes of the Fourier transforms of a complex signal. This is because measuring magnitude (e.g., by aggregating energy over time) is much easier than measuring phase (which requires detecting rapid changes). We refer the reader to the survey articles [37, 26] for more details about the theory and applications of phase retrieval.

In this paper, we focus on the real-valued generalized phase retrieval problem, where the Fourier transform is replaced by a general linear operator. We first give a formal definition of this problem.

**Definition 1.1** (Phase Retrieval). Let $x \in \mathbb{R}^d$ be the ground-truth vector. Let $a_1 \ldots a_n \in \mathbb{R}^d$ be $n$ sampling vectors and let $y_i = \langle a_i, x \rangle^2 \in \mathbb{R}$ be the corresponding intensity measurements. Given $(a_i, y_i)_{i=1}^n$ as input, the task is to recover $x$.

Note that it is impossible to distinguish between $x$ and $-x$, so it is sufficient to recover either one. Under certain assumptions (e.g., when the $a_i$'s are Gaussian distributed), the phase retrieval problem in Definition 1.1 can be solved in polynomial time with provable recovery guarantees. This was first achieved via approaches based on semidefinite programming (SDP) relaxations (see, e.g., Candès et al. [5]). In practice, the problem is often solved using first-order optimization algorithms such as gradient descent. It is well-established that, although many natural formulations of phase retrieval have nonconvex objectives, all local optima are globally optimal under certain assumptions [34, 3, 40]. An example of such objective function is the following:

$$\text{minimize} \quad f(z) = \sum_{i=1}^n (y_i - \langle a_i, z \rangle^2)^2 \quad \text{subject to} \quad z \in \mathbb{R}^d.$$

Submitted to 38th Conference on Neural Information Processing Systems (NeurIPS 2024). Do not distribute.

However, existing analyses of such landscape results often rely on strong assumptions, such as the sampling vectors $a_i$'s are i.i.d. Gaussian. Our work is motivated by the following questions: Can we relax the assumptions used in proving landscape results in many tractable nonconvex problems? In the context of phase retrieval, what happens if a small fraction of the $(a_i, y_i)$'s are changed adversarially? We focus on the following strong contamination model (see, e.g., [13]).

**Definition 1.2** ($\epsilon$-Corruption). An algorithm first specifies the number of samples $n$, and $n$ samples are drawn independently from some unknown distribution $D$. The adversary is allowed to replace up to $\epsilon n$ samples with arbitrary points. The modified set of $n$ samples is then given to the algorithm as input. We say that a set of samples is $\epsilon$-corrupted if it is generated by the above process. [1]

Under the $\epsilon$-corruption model for high-dimensional data, a common goal is to design efficient algorithms that can achieve dimension-independent error guarantees. Early work in robust statistics [42, 23, 25] provided sample-efficient estimators for various tasks, but with runtimes exponential in the dimension. A recent line of work, initiated by [13, 28], has developed computationally efficient robust algorithms for many fundamental high-dimensional tasks. There has been significant progress in the algorithmic aspects of robust high-dimensional statistics (see, e.g., [12]).

We now formally define the main problem that we pose and study in this paper.

**Problem 1.3** (Outlier-Robust Phase Retrieval). Let $\epsilon > 0$. Let $x \in \mathbb{R}^d$ be the ground-truth vector with $\|x\|_2 = 1$. First, $n$ sampling vectors $(a_i)_{i=1}^n$ are drawn i.i.d. from $\mathcal{N}(0, I) \in \mathbb{R}^d$. Let $y_i = \langle a_i, x \rangle^2$ be the corresponding intensity measurements. Then, an adversary arbitrarily corrupts an $\epsilon$-fraction of the $(a_i, y_i)$'s. Finally, the corrupted $(a_i, y_i)$'s are given to the algorithm as input. The task is to find a vector $z \in \mathbb{R}^d$ such that $\min\{\|z - x\|_2, \|z + x\|_2\} \leq \Delta$ for some precision parameter $\Delta > 0$.

Note that we allow corruption in both the sampling vectors $a_i \in \mathbb{R}^d$ and the intensity measurements $y_i \in \mathbb{R}$. We would like to answer the following algorithmic question:

> *Can we design a provably robust and near sample-optimal algorithm for the $\epsilon$-corrupted phase retrieval problem (Problem 1.3) that runs in nearly-linear time?*

## 1.1 Our Results and Contributions

In this paper, we answer the above question affirmatively. We first state the main result of our paper.

**Theorem 1.4** (Main, Informal). *Consider the outlier-robust phase retrieval problem (Problem 1.3). Let $\Delta > 0$. Given an $\epsilon$-corrupted set of $n = \widetilde{\Omega}(d \log^2(1/\Delta))$ samples, we can compute $z \in \mathbb{R}^d$ in time $\widetilde{O}(nd)$ such that $\min(\|z - x\|_2, \|z + x\|_2) \leq \Delta$ with probability at least $0.8$.*

Our algorithm has near-optimal sample complexity, because even without corruption, recovering the ground-truth vector $x$ in general requires $\Omega(d)$ samples because there are $d$ degrees of freedom in $x$. Moreover, our algorithm runs in time nearly-linear in the size of the input, and provably recovers the ground-truth vector $x$ with arbitrary precision $\Delta$. The formal version of Theorem 1.4 is stated as Theorem 3.1 in Section 3.

We remark that the success probability of Theorem 1.4 can be boosted to $1 - \delta$ for any $\delta > 0$ by incurring an additional factor of $T = O(\log(1/\delta))$ in the sample complexity and runtime. We can randomly partition the input into $T$ equal-sized disjoint sets and run our algorithm on each set to obtain $T$ solutions $Z = \{z_1, \ldots, z_T\}$. If we output a solution $z^\star$ that has the maximum number of points in $Z$ within distance $2\Delta$, we can show that $r(z^\star) \leq 3\Delta$ with probability at least $1 - \delta$.

Our main conceptual contribution is to propose and study the outlier-robust phase retrieval problem, where a small fraction of the input data is adversarially corrupted. Note that we allow arbitrary corruption in both the sampling vectors $a_i \in \mathbb{R}^d$ and the intensity measurements $y_i \in \mathbb{R}$. The nonconvex optimization landscape of phase retrieval is well understood when the $a_i$'s are Gaussian distributed, but the adversarial robustness of such landscape results is largely unexplored.

Our main technical contributions include the design and analysis of a near sample-optimal and nearly-linear time algorithm that provably solves the phase retrieval problem in the presence of outliers. Our approach provides a conceptually simple two-step framework for developing outlier-robust algorithms for tractable nonconvex problems that combines the robustness of spectral initialization and the efficiency of the subsequent robust gradient descent.

---

[1]We write $G \subseteq [n]$ for the (remaining) good samples, and $B = [n] \setminus G$ for the corrupted samples.

## 1.2 Our Approach and Techniques

When there are infinite samples and no corruption, the objective function $f(z)$ can be simplified as

$$f(z) = \mathop{\mathbf{E}}_{a \sim \mathcal{N}(0, I_d)} \left[ (\langle a_i, z \rangle^2 - y_i)^2 \right] = 3 \|x\|_2^4 + 3 \|z\|_2^4 - 2 \|x\|_2^2 \|z\|_2^2 - 4 \langle x, z \rangle^2 . \tag{1}$$

Even though $f(z)$ is nonconvex, we know that it has no spurious local optima [34, 3, 40].

Our approach follows the general structure of Candès et al. [3], which uses a two-step procedure. The first step uses spectral techniques to find an initial guess that is close enough to the ground truth. The second step applies gradient descent to converge to the final solution. However, both steps are susceptible to adversarial corruption. We develop nearly-linear time and provably robust algorithms for both steps and combine them to get our main result.

**Step 1: Robust Spectral Initialization.** When there is no adversarial corruption, the empirical second-moment matrix $Y = (1/n) \sum_{i=1}^{n} y_i a_i a_i^\top$ has expectation $\mathbf{E}[Y] = I + 2xx^\top$, so its top eigenvector is close to $x$. However, the adversary can arbitrarily change the top eigenvector.

To circumvent this issue, we assign a (nonnegative) weight $w_i$ to each sample, and let $Y_w$ denote the weighted intensity-based second-moment matrix $Y_w = \sum_{i=1}^{n} w_i y_i a_i a_i^\top$. Ideally, if the weights $w$ are uniformly distributed on the remaining clean samples, the top eigenvector of $Y_w$ will align with $x$. We propose a novel optimization problem that can be used to find a weighting $w$ such that $Y_w$ must be close to the unknown unbiased expectation $I + 2xx^\top$. Moreover, we show that such a weight $w$ can be computed in nearly-linear time.

**Step 2: Approximate Gradient Descent.** Starting with the initial guess $z_1 \in \mathbb{R}^d$ produced by the robust spectral initialization, we want to apply gradient descent to recover the ground truth $x \in \mathbb{R}^d$. Without corruption, if the initialization is close enough to $x$, each iteration will bring $z$ closer to $x$ by a constant factor. This convergence guarantee can be compromised by the corrupted samples.

At a high level, approximating the gradient at a specific point amounts to a robust mean estimation problem (for the underlying distribution of the gradients). When the input data is $\epsilon$-corrupted, the gradients of the $n$ samples can be viewed as an $\epsilon$-corrupted set of vectors. We can approximate the true gradient using this $\epsilon$-corrupted set of $n$ gradients using robust mean estimation algorithms.

## 1.3 Related and Prior Works

**Phase Retrieval.** The problem of phase retrieval arises in many areas of science and engineering [11, 33]. Early research on this problem proposes error-reduction algorithms [22, 17, 18]. Convex and nonconvex optimization with various objective functions were later proposed and achieved exact recovery [43, 3–5, 38]. Follow-up works generalize to robust phase retrieval where the observations are subject to perturbations [45, 27, 7, 6, 31].

**Nonconvex Optimization.** Even though optimizing a nonconvex function is NP-Hard in general, recent works showed that many nonconvex functions are locally optimizable due to discrete or rotational symmetry. Besides phase retrieval, it is known that all local optima are globally optimal for natural nonconvex formulations of a wide range of machine learning problems, such as matrix completion [21], matrix sensing [2], phase synchronization [1], dictionary learning [39], and tensor decomposition [20] (see also Chapter 7 of [44]). Closely related to our work, a recent line of work explored the robustness of these landscape results: [30] studied matrix sensing in the $\epsilon$-corrupted model and [8, 19] studied matrix completion and matrix sensing in semi-random models.

**High-Dimensional Robust Statistics.** Recent works in high-dimensional robust statistics developed nearly-linear time algorithms for the problem of robust mean estimation [9, 16, 29]. Prior works [35, 14] developed meta-algorithms for finding *first-order* stationary points with dimension-independent accuracy guarantees, which is closely related to the robust gradient descent procedure that we use.

## 1.4 Roadmap

We first introduce notations and background in Section 2. Then we give an overview of our approach in Section 3. Next, we focus on how to get an initialization that is close enough to the ground truth $x$ in Section 4. After the initialization, we use robust mean algorithms to estimate gradients to converge to the desired accuracy in Section 5. Finally, we conclude in Section 6 and discuss open problems.

## 2 Preliminary and Background

**Notation.** We write $[n]$ for the set of integers $\{1, \ldots, n\}$. We use $\{e_1, \ldots, e_d\}$ for the standard unit vector basis in $\mathbb{R}^d$ and $I$ for the identity matrix. For a vector $x$, we denote its $\ell_1$, $\ell_2$ and $\ell_\infty$ norm as $\|x\|_1$, $\|x\|_2$ and $\|x\|_\infty$, respectively, and write the $i^{\text{th}}$ coordinate in $x$ as $x_i$. For vectors $x, y \in \mathbb{R}^d$, we denote its inner product as $\langle x, y \rangle = x^\top y$. For a matrix $A$, we use $\|A\|_2$, $\|A\|_*$, and $\|A\|_F$ as its operator norm, nuclear norm, and Frobenius norm, respectively. We write $\lambda_k(A)$ as the $k$th-largest eigenvalues of $A$, and $\overline{\lambda}_k(A)$ as the sum of the $k$ largest eigenvalues. A symmetric $n \times n$ matrix $A$ is said to be positive semidefinite (PSD) if for all vectors $x \in \mathbb{R}^n$, $x^\top A x \geq 0$. For two symmetric matrices $A$ and $B$, we write $A \preceq B$ when $B - A$ is positive semidefinite.

**Packing SDP.** We will use nearly-linear time solvers for the following packing SDP.

$$\max_w \ \|w\|_1 \quad \text{subject to} \quad \sum_{i=1}^n w_i A_i \preceq I, \quad \overline{\lambda}_k \left( \sum_{i=1}^n w_i B_i \right) \leq k, \quad w_i \geq 0, \forall i. \quad (*)$$

**Lemma 2.1** ([10]). *Given an instance of optimization* (*) *with semi-positive definite matrices $A_i \in \mathbb{R}^{d_1 \times d_1}$ and $B_i \in \mathbb{R}^{d_2 \times d_2}$ with $A_i = C_i C_i^\top$, $B_i = D_i D_i^\top$ for all $i = 1, 2, \cdots, m$, together with integer $k > 0$, error tolerance $\epsilon_0 \geq 1/m^2$, and failure probability $\delta_0$, there is an algorithm that runs in time $\widetilde{O}((t_C + t_D + d_1 + d_2) \operatorname{poly}(1/\epsilon_0, \log 1/\delta_0))$, where $t_{C_i}$ and $t_{D_i}$ are the time take to perform a matrix product with $C_i$ and $D_i$ respectively and $t_C = \sum_{i=1}^n t_{C_i}$ and $t_D = \sum_{i=1}^n t_{D_i}$, and outputs $w'$ with $\|w'\|_1 \geq (1 - \epsilon_0)\mathsf{OPT}$ where $\mathsf{OPT}$ is optimal value, with probability at least $1 - \delta_0$.*

**Computing the Top Eigenvector.** We use power method to compute the top eigenvector of a matrix.

**Lemma 2.2** (Power Method for Top Eigenvector, e.g., [41]). *Let $A \in \mathbb{R}^{d \times d}$ and let $\lambda_1$ be its largest eigenvalue. For any $\overline{\delta} \in (0, 1)$, there exists an algorithm that takes $A$ and outputs a unit vector $x \in \mathbb{R}^d$ in time $O(t \log(d)/\overline{\delta})$ such that $x^T A x \geq (1 - \overline{\delta})\lambda_1$ with probability at least $0.99$, where $t$ is the time required to compute $Av$ for an arbitrary $v \in \mathbb{R}^d$.*

**Robust Mean Estimation.** Another tool we use is robust mean estimation in the $\epsilon$-corruption model for distributions with bounded covariance. We use robust mean estimation algorithms to approximate the true gradient under adversarial corruption.

**Lemma 2.3** (Robust Mean Estimation, e.g., [15]). *Let $\mathcal{D}$ be a distribution on $\mathbb{R}^d$ with unknown mean $\mu$ and unknown covariance matrix $\Sigma$ where $\Sigma \preceq \sigma^2 I$. Let $\epsilon_0$ be a sufficiently small universal constant. Let $0 < \epsilon < \epsilon_0$ and $\delta > 0$. Given an $\epsilon$-corrupted set of $n$ samples drawn from $\mathcal{D}$, we can output a vector $\widehat{\mu} \in \mathbb{R}^d$ in time $\widetilde{O}(nd \log(1/\delta))$ such that, with probability at least $1 - \delta - \exp(-n\epsilon)$, we have $\|\widehat{\mu} - \mu\|_2 = O\left( \sqrt{\epsilon} + \sqrt{\frac{d}{n\delta}} + \sqrt{\frac{d(\log d + \log 1/\delta)}{n}} \right) \sigma.$*

## 3 Overview

We first state a formal version of our main result.

**Theorem 3.1** (Main). *Consider the setting of Problem 1.3. Let $0 < \epsilon < \epsilon'$ for some universal constant $\epsilon'$ and let $\Delta > 0$. Given an $\epsilon$-corrupted set of $n = \widetilde{\Omega}(d \log^2(1/\Delta))$ samples, we can compute a vector $z \in \mathbb{R}^d$ in time $\widetilde{O}(nd \log(1/\Delta))$ such that $r(z) = \min\{\|z - x\|_2, \|z + x\|_2\} \leq \Delta$ with probability at least $0.8$.*

Theorem 3.1 requires two key technical lemmas: the robust spectral initialization (Lemma 3.2) and the approximate gradient descent (Lemma 3.3).

We first show that the spectral initialization can be done in nearly linear time with high probability, the proof of which can be found in Section 4.

**Lemma 3.2** (Robust Spectral Initialization). *Under the setting of Problem 1.3, for any $0 < \epsilon < \epsilon'$ for some universal constant $\epsilon' > 0$, given an $\epsilon$-corrupted set of $n = \widetilde{\Omega}(d)$ samples, we can compute a vector $z_0 \in \mathbb{R}^d$ of the ground truth $x$ in time $\widetilde{O}(nd)$ such that $r(z_1) = \min\{\|z_1 - x\|_2, \|z_1 + x\|_2\} \leq \frac{1}{8}$ with probability at least $0.95$.*

Then, with such initialization results, we can proceed to show that an approximate gradient descent algorithm can be used to find an arbitrary approximation of the ground truth in Section 5.

**Lemma 3.3** (Robust Gradient Descent). *Consider the setting of Problem 1.3. Let $\Delta > 0$ be the desired precision. Let $0 < \epsilon < \epsilon_0$ for some sufficiently small universal constant $\epsilon_0$. Given an $\epsilon$-corrupted set of $n = \widetilde{\Omega}(d \log^2(1/\Delta))$ samples and an initial guess $z_1$ such that $r(z_1) = \min(\|z_1 - x\|_2, \|z_1 + x\|_2) \leq 1/8$, we can compute a vector $z \in \mathbb{R}^d$ in time $\widetilde{O}(nd)$ such that $r(z) \leq \Delta$ with probability at least $0.95$.*

For technical reasons, we cannot use the same set of samples for both the robust spectral initialization and the approximate gradient descent. Therefore, we partition the $\epsilon$-corrupted set of $2n$ samples into two equally sized disjoint sets, using one set for each algorithm.

*Proof of Theorem 3.1.* Let $2n = \widetilde{\Omega}(d \log^2(1/\Delta))$ be a set of $\epsilon/2$-corrupted samples. We partition the input into two disjoint sets of $n$ samples. Both sets are $\epsilon$-corrupted. By Lemmas 3.2 and 3.3, for any $\epsilon \in [0, \epsilon']$ and $\Delta > 0$, our algorithm takes the first set of samples and output a vector $z'$ in time $\widetilde{O}(nd)$ such that $r(z') \leq 1/8$ with probability at least $0.95$. Then, using $z'$ and the second set of samples, our algorithm can output $z \in \mathbb{R}^d$ in time $\widetilde{O}(nd)$ such that $r(z) \leq \Delta$ with probability at least $0.95$. The overall success probability is at least $0.8$, and the combined running time is $\widetilde{O}(nd)$. $\quad\square$

## 4 Robust Spectral Initialization

We dedicate this section to proving Lemma 3.2: Given an $\epsilon$-corrupted set of $(a_i, y_i)$'s, we can compute an initial guess $z_1 \in \mathbb{R}^d$ that is close to the ground truth $x$, where $\min(\|z_1 - x\|_2, \|z_1 + x\|_2) \leq 1/8$. To build some intuition, consider the following intensity-based covariance matrix $Y = \frac{1}{n} \sum_{i=1}^n y_i a_i a_i^\top$, where each $a_i$ is drawn independently from $\mathcal{N}(0, I)$ and $y_i = \langle a_i, x \rangle^2$. The expectation of this matrix is $\mathbb{E}[Y] = I + 2xx^\top$. In other words, when there are enough samples and no adversarial corruption, we can obtain a good guess of the ground truth $x$ (or $-x$) by computing the top eigenvector of $Y$. However, we cannot rely on this approach in adversarial settings.

To tackle this issue, we propose a nearly-linear time preprocessing step (Algorithm 1) that can recover the true expectation of $Y$ under adversarial corruptions. Algorithm 1 assigns a non-negative weight to each sample. For a weight vector $w \in \mathbb{R}^n$ and a set of indices $S \subseteq [n]$, the weighted intensity-based covariance matrix is defined as $Y_{S,w} \doteq \sum_{i \in S} w_i y_i a_i a_i^\top$, and we omit $S$ when $S = [n]$. The feasible region for the weight vector is: $\Delta_{n,\epsilon} := \left\{ w \in \mathbb{R}^n : \|w\|_1 = 1 \text{ and } \forall i \in [n], 0 \leq w_i \leq \frac{1}{(1-\epsilon)n} \right\}$.

A weight $w$ defines an empirical distributions over the samples $(a_i, y_i)_{i=1}^n$, where the largest probability assigned to any point is $\frac{1}{(1-\epsilon)n}$. Ideally, we would like to find a weight vector $w^* \in \Delta_{n,\epsilon}$ that assigns its weight uniformly to all the uncorrupted samples, i.e., $w_i^* = \frac{1}{(1-\epsilon)n} \cdot \mathbb{1}_{i \in G}$. To find a suitable weighting $w$, we use the following optimization problem (**) in which $\overline{\lambda}_2$ returns the sum of the top two eigenvalues (commonly known as the Ky Fan $k$ norm for $k = 2$).

$$\min_w \quad \overline{\lambda}_2 \left( \sum_{i=1}^n w_i y_i a_i a_i^\top \right) \qquad \text{subject to} \quad 0 \leq w_i \leq \frac{1}{(1-\epsilon)n}, \forall i \in [n], \quad \sum_{i=1}^n w_i = 1. \quad (**)$$

At a high level, our main observation is that $y_i a_i a_i^\top$ is always a positive semidefinite matrix as $y_i \geq 0$. Consequently, the adversary can only *add* extra directions with large eigenvalues, but will not be able to remove the eigendirection of $x$. By minimizing the Ky Fan 2 norm, we can remove any directions added by the adversary and make sure that the only remaining large eigendirection is close to $x$.

Let $\delta \geq 0$ be some constant to be determined. We show that we can obtain a robust spectral initialization by solving the packing SDP problem (*), which can be solved efficiently using Lemma 2.1. In particular, to fit the reweighting problem of (**) into the framework of the generalized packing problem (*), we define the following constraint matrices for all $i \in [n]$ :

$$A_i := (1 - \epsilon)n \cdot e_i e_i^\top, \qquad B_i := \tfrac{1}{2}(1 - \delta) y_i a_i a_i^\top. \qquad (2)$$

The matrices $(A_i)_{i=1}^n$ are used to implement the constraint that $w \in \Delta_{n,\epsilon}$. The matrices $(B_i)_{i=1}^n$ help make sure the sum of the top two eigenvalues of $Y_w$ must be at most roughly 4, because $\overline{\lambda}_2(Y_{w^*}) \approx 4$.

---
**Algorithm 1** Robust Spectral Initialization
---
    **Input:** $\epsilon$-corrupted samples $(a_i, y_i)_{i \in [n]}$
    **Output:** The initial guess $z' \in \mathbb{R}^d$
 1: **function** ROBUSTINIT($\{(a_i, y_i)\}_{i \in [n]}$)
 2:    $\{A_i, B_i\} \leftarrow$ Constraint matrices as defined in Equation (2)
 3:    $w'' \leftarrow$ Solution to Optimization (*) with constraints $\{A_i, B_i\}$, $k = 2$, and precision $\epsilon_0 = 0.9$
      as in Lemma 2.1
 4:    $w' \leftarrow w'' / \|w''\|_1$
 5:    $z_1 \leftarrow$ TOPEIGENVECTOR($Y_{w'}$) as in Lemma 2.2 with sufficiently small constant $\bar{\delta}$.
 6:    **return** $z_1$
 7: **end function**
---

First, we show that the weight $w'$ computed by Algorithm 1 can ensure the weighted intensity-based covariance matrix $Y_{w'}$ is close enough to the unbiased expectation $I + 2xx^\top$.

**Lemma 4.1.** *With probability at least* $0.98$*, the* $w'$ *outputted by Algorithm 1 satisfies:*

$$\left\| Y_{w'} - (I + 2xx^\top) \right\|_2 = O(\delta) \tag{3}$$

In order to show Lemma 4.1, we need the following auxiliary Lemma 4.2, the proof of which can be found in Section A. Intuitively, Lemma 4.2 suggests that any weight $w$ in the feasible region $\Delta_{n,2\epsilon}$ will not have a huge impact on the properties of uncorrupted measurements.

**Lemma 4.2.** *For any* $\delta_0 > 0$*, and sufficiently small* $\epsilon \geq 0$*, given a set of* $n$ $\epsilon$*-corrupted samples with* $n > \widetilde{\Omega}(d)$*, with probability at least* $0.98$*, we have* $\left\| Y_{G,w} - (I + 2xx^\top) \right\|_2 \leq \delta_0$ *for all* $w \in \Delta_{n,2\epsilon}$*.*

Using Lemma 4.2, we provide a proof sketch for Lemma 4.1, and defer the details to Section A.

*Proof.* We condition on the fact that the event of Lemma 4.2 holds (with probability at least $0.98$) for $\delta_0 = \delta$. Thus, for the remaining of the proof, we assume that for all $w \in \Delta_{n,2\epsilon}$, it holds that $\left\| Y_{G,w} - (I + 2xx^\top) \right\|_2 \leq \delta$.

Let $\lambda_1$ and $\lambda_2$ be the top two eigenvalues of $Y_{w'}$, with $v_1$ and $v_2$ to be their corresponding eigenvectors.

Note that the largest eigenvalue of $I + 2xx^\top$ is $3$, and the rest of the eigenvalues are all $1$. In the proof, we show that the eigenvalues of $Y_{w'}$ are also close to the ones of $I + 2xx^\top$. Our proof consists of two parts. We first establish lower bounds for $\lambda_1$ and $\lambda_2$, and then find an upper bound for $\lambda_1 + \lambda_2$.

**Lower Bound.** Since $y_i a_i a_i^\top \succeq 0$ for any $i \in [n]$, for any positive weight vector $w \in \Delta_{n,\epsilon}$, we have $Y_{G,w} \preceq Y_w$. Thus a lower bound on eigenvalues of $Y_{G,w'}$ will also be a lower bound on $Y_{w'}$.

For the top eigenvalue $\lambda_1$ of $Y_{w'}$, it holds

$$\lambda_1 = v_1^\top Y_{w'} v_1 \geq x^\top Y_{w'} x \geq x^\top Y_{G,w'} x \geq x^\top (I + 2xx^\top) x - \delta = 3 - \delta. \tag{4}$$

Similarly, for the second largest eigenvalue $\lambda_2$ of $Y_{w'}$, we have:

$$\lambda_2 = v_2^\top Y_{w'} v_2 \geq v_2^\top Y_{G,w'} v_2 \geq v_2^\top (I + 2xx^\top) v_2 - \delta = 1 + 2 \langle v_2, x \rangle^2 - \delta \geq 1 - \delta. \tag{5}$$

**Upper Bound.** Through the optimization problem (*), a weight $w''$ is calculated such that $Y_{w''}$ are operator-norm upper-bounded by the constraint parameters. Let OPT be the value of the optimal solution of the optimization problem (*). The desired uniform weight vector over the good samples $w^* \in \Delta_{n,2\epsilon}$ is also a feasible solution to this optimization problem because $Y_{w^*}$ satisfy the optimization constraints due to Lemma 4.2. Since $w''$ is an $\epsilon_0$-approximation to the problem, we have

$$\|w''\|_1 \geq (1 - \epsilon_0)\text{OPT} \geq (1 - \epsilon_0) \|w^*\|_1 = 1 - \epsilon_0$$

By optimization constraints, the Ky Fan 2-norm of $\sum_i w_i'' B_i = \frac{1}{2}(1 - \delta) Y_{w''} \leq 2$, and consequently,

$$\lambda_1 + \lambda_2 = \frac{1}{\|w''\|_1} \overline{\lambda}_2(Y_{w''}) \leq \frac{4}{(1-\epsilon_0)(1-\delta)} \ . \tag{6}$$

By combining inequalities (4), (5), and (6), we have shown that the top two eigenvalues of $Y_{w'}$ are close to $3$ and $1$. Since the rest of the eigenvalues of $Y_{w'}$ can also be bounded, we can conclude that $\left\| Y_{w'} - (I + 2xx^\top) \right\|_2 = O(\delta)$. $\qquad\square$

We can now show the closeness between the top eigenvector of $Y_{w'}$ and the ground truth.

**Lemma 4.3.** *There exists an universal constant $\epsilon'$ such that if $0 \leq \epsilon \leq \epsilon'$, and Algorithm 1 receives in input an $\epsilon$-corrupted set of samples, then it outputs $z_1 \in \mathbb{R}^d$ such that with probability at least $0.95$ it holds $r(z_1) \leq \frac{1}{8}$.*

*Proof.* We condition on the fact that the event of Lemma 4.1 holds (with probability at least $0.98$). Let the eigendecomposition of $Y_{w'}$ be $Y_{w'} = \sum_{i \in [d]} \lambda_i v_i v_i^\top$, where $\lambda_1 \geq \ldots \geq \lambda_d$. Under the basis $\{v_1, \cdots, v_d\}$, the ground truth $x$ can be represented as $x = \sum_{i \in [d]} \alpha_i v_i$. Note that $\|x\|_2^2 = \sum_{i \in [d]} \alpha_i^2 = 1$. By Lemma 4.1, we have $\left\| Y_{w'} - (I + 2xx^\top) \right\|_2 = O(\delta)$. Thus, we have

$$x^\top Y_{w'} x \geq 3 - O(\delta) \quad \text{and}$$
$$x^\top Y_{w'} x \leq \lambda_1 \alpha_1^2 + \lambda_2(1 - \alpha_1^2) \leq (3 + O(\delta))\alpha_1^2 + (1 + O(\delta))(1 - \alpha_1^2) \leq 1 + 2\alpha^2 + O(\delta).$$

This implies $\alpha_1^2 \geq 1 - O(\delta)$. As a result,

$$r^2(v_1) = \left( \min\{\|v_1 - x\|_2^2, \|v_1 + x\|_2^2\} \right) = \min\{(\alpha_1 - 1)^2, (\alpha_1 + 1)^2\} + \sum_{i=2}^d \alpha_i^2$$
$$= \min\{2 - 2\alpha_1, 2 + 2\alpha_1\} = O(\delta).$$

The last inequality holds as long as $\delta$ is sufficiently small. Let $z_1 = \sum_{i \in [d]} \beta_i v_i$ be the unit vector approximating $v_1$ returned by the algorithm. By Lemma 2.2, we have that $z_1^\top Y_{w'} z_1 \geq (1 - \overline{\delta})\lambda_1$ with probability at least $0.99$. Thus, we have:

$$z_1^\top Y_{w'} z_1 \geq (1 - \overline{\delta})\lambda_1 \geq (1 - \overline{\delta})(3 - O(\delta)) \geq 3 - O(\delta + \overline{\delta}) \quad \text{and}$$
$$z_1^\top Y_{w'} z_1 \leq \lambda_1 \alpha_1^2 + \lambda_2(1 - \alpha_1^2) \leq 1 + 2\beta_1^2 + O(\delta).$$

Again, this implies that $\beta_1^2 \geq 1 - O(\delta + \overline{\delta})$. We can show that $\min\{\|v_1 - z_1\|_2^2, \|v_1 + z_1\|_2^2\} = O(\overline{\delta} + \delta)$. By the triangle inequality, we can conclude that $r^2(z_1) = O(\overline{\delta} + \delta) \leq 1/64$, where the last inequality is obtained by choosing sufficiently small $\delta$ and $\overline{\delta}$. Therefore, there exists an universal constant $\epsilon' \geq 0$ such that for all $0 \leq \epsilon \leq \epsilon'$, Algorithm 1 takes $n = \widetilde{\Omega}(d)$ samples and outputs $z_1$ such that $r(z_1) \leq 1/8$ with probability at least $0.95$. $\qquad \square$

**Lemma 4.4.** *Algorithm 1 runs in time $\widetilde{O}(nd)$.*

*Proof of Lemma 4.4.* Since we have the factorization of the rank-two matrices $A_i$ and rank-one matrices $B_i$ for all $i = 1, 2, \ldots, n$, and the time to perform a matrix-vector product with $C_i$ and $D_i$ is $O(d)$. Therefore, by Lemma 2.1, with $t_C$ and $t_D$ to be $\widetilde{O}(nd)$, Line 3 runs in $\widetilde{O}(nd)$ time. In Line 5, by Lemma 2.2, the top eigenvector of $Y_{w'}$ can be computed in $\widetilde{O}(n \log d)$ time using power method. Scaling in Line 4 runs in $O(n)$ time. As a result, Algorithm 1 runs in $\widetilde{O}(nd)$ time. $\qquad \square$

We can directly combine Lemma 4.3 and Lemma 4.4 to finish the proof of Lemma 3.2.

# 5 Robust Gradient Descent

After the robust spectral initialization in Section 4, we have an initial guess $z_1 \in \mathbb{R}^d$ that is close to the ground truth $x$ or $-x$. Without loss of generality, we can assume that $z_1$ is closer to $x$ than to $-x$. In this section, we prove Lemma 3.3: Given an initial guess $z_1$ with $\|z_1 - x\|_2 \leq 1/8$, we can use a robust gradient descent algorithm (Algorithm 2) to recover $x$ to any desire precision $\Delta > 0$. It is well-known that gradient descent can achieve geometric convergence rates in non-adversarial settings. We show that Algorithm 2 achieves a similar convergence rate even when the input is $\epsilon$-corrupted.

Consider the natural nonconvex formulation: $\min_{z \in \mathbb{R}^d} \sum_{i=1}^n f_i(z)$ where $f_i(z) \doteq \left( \langle a_i, z \rangle^2 - y_i \right)^2$. Let $g_i$ denote the gradient of $f_i$ with respect to $z$. Let $\mathcal{D}_z$ denote the distribution of $g_i(z) \in \mathbb{R}^d$ when there is no adversarial corruption. Formally, $g(z) \sim \mathcal{D}_z$ is distributed as

$$g(z) = \frac{\partial}{\partial z}\left[ \left( \langle a, z \rangle^2 - \langle a, x \rangle^2 \right)^2 \right] = -4\left( \langle a, z \rangle^2 - \langle a, x \rangle^2 \right) \langle a, z \rangle a \quad \text{where} \quad a \sim \mathcal{N}(0, I) . \quad (7)$$

To run gradient descent, we want to estimate the *expected true gradient* $\mu_z \doteq \mathbb{E}\, g(z)$ using samples. The challenge is that the input samples $\{(a_i, y_i)\}_{i \in [n]}$ are $\epsilon$-corrupted, and consequently the gradients $\{g_i(z)\}_{i \in [n]}$ is an $\epsilon$-corrupted set of vectors drawn from $\mathcal{D}_z$. To address this, we use robust mean estimation algorithms (e.g., [16]) to approximate $\mu_z$, the true mean of $\mathcal{D}_z$.

---

**Algorithm 2** Robust Gradient Descent

---

    **Input:** $\epsilon > 0$, an $\epsilon$-corrupted set of $n$ samples $\{(a_i, y_i)\}_{i \in [n]}$, initial guess $z_1 \in \mathbb{R}^d$ with $\|z_1 - x\| \le 1/8$, and desired precision $\Delta > 0$.

    **Output:** $z \in \mathbb{R}^d$ such that $\|z - x\|_2 \le \Delta$ where $x$ is the ground truth.

1: **procedure** ROBUSTGD($\epsilon, \{(a_i, y_i)\}_{i \in [n]}, z_1, \Delta$)
2:     $T \leftarrow O(\log(1/\Delta))$, $\eta \leftarrow 1/300$
3:     $\{N_1, N_2, \cdots, N_T\} \leftarrow$ a random disjoint partition of $[n]$ such that $|N_t| = n/T$ for all $t \in [T]$
4:     **for** $t = 1, 2, \ldots, T$ **do**
5:         $\widehat{\mu}_{z_t} \leftarrow$ Robust mean estimation on input $\{g_i(z_t)\}_{i \in N_t}$ using Lemma 5.2
6:         $z_{t+1} \leftarrow z_t - \eta\, \widehat{\mu}_{z_t}$
7:     **end for**
8:     **return** $z_{T+1}$
9: **end procedure**

---

The error guarantee of robust mean estimation algorithms depends on the covariance matrix $\Sigma_z$ of the distribution $\mathcal{D}_z$. The next lemma upper bounds the spectral norm of $\Sigma_z$.

**Lemma 5.1.** *Let $\mathcal{D}_z$ be the distribution of gradients at $z$ as defined in Equation* (7). *For any $z \in \mathbb{R}^d$ with $\|z - x\|_2 \le 1$, the covariance matrix $\Sigma_z$ of $\mathcal{D}_z$ satisfies $\Sigma_z \preceq O(\|z - x\|_2^2)I$.*

The proof of Lemma 5.1 is deferred to Appendix B. Given Lemma 5.1, we can show that robust mean estimation algorithms can approximate $\mu_z$ with small error. For technical reasons, we randomly partition the input samples $(a_i, y_i)$ into $T$ subsets, and use one subset in each iteration. With high probability, each partition has at most $(2\epsilon)$-fraction of corrupted samples

**Lemma 5.2.** *Consider any $z \in \mathbb{R}^d$ with $\|z - x\|_2 \le 1$. Let $\mu_z$ be the mean of $\mathcal{D}_z$ as defined in Equation* (7). *Fix universal constants $c > 0$ and $\epsilon_0 = \Theta(c^2)$. Let $2\epsilon < \epsilon_0$ and $\delta > 0$. Given a $(2\epsilon)$-corrupted set of $m = \Omega(d \log d/\delta)$ samples drawn from $\mathcal{D}_z$, we can compute $\widehat{\mu}_z$ in time $\widetilde{O}(md \log(1/\delta))$ such that $\|\widehat{\mu}_z - \mu_z\|_2 \le c\, \|z - x\|_2$ with probability at least $1 - O(\delta)$.*

*Proof of Lemma 5.2.* Since $2\epsilon < \epsilon_0$, we can view the $(2\epsilon)$-corrupted set of $m$ samples as $\epsilon_0$-corrupted. We need to replace $2\epsilon$ with $\epsilon_0$ to reduce the failure probability of Lemma 2.3. This weakens the error guarantee of Lemma 2.3, but the resulting $\widehat{\mu}_z$ is still accurate enough for our algorithm.

We apply Lemma 2.3 to the $\epsilon_0$-corrupted set of $m$ vectors drawn from $\mathcal{D}_z$. By Lemma 5.1, the covariance matrix of $\mathcal{D}_z$ satisfies $\Sigma_z \preceq O(\|z - x\|_2^2)I$. Consequently, for sufficiently large $m = \Theta(d \log d/\delta)$ and sufficiently small $\epsilon_0 = O(c^2)$, the error guarantee of Lemma 2.3 is $O\left(\sqrt{\epsilon_0} + \sqrt{\frac{d}{m\delta}} + \sqrt{\frac{d(\log d + \log(1/\delta))}{m}}\right) \|z - x\|_2 \le c\, \|z - x\|_2$. The success probability is at least $1 - \delta - \exp(-\epsilon_0 m) = 1 - O(\delta)$. $\qquad\square$

Lemma 5.2 shows that even with a $(2\epsilon)$-corrupted set of gradients, the true gradient $\mu_z$ can be estimated up to an additive error proportional to the distance between $z$ and $x$. The next lemma shows that such an approximate gradient is sufficient for gradient descent to converge, reducing the distance to the ground truth $x$ by a constant factor in each iteration.

**Lemma 5.3.** *Suppose in iteration $t$ of Algorithm 2, the current solution $z_t$ satisfies $\|z_t - x\|_2 \le 1/8$, and the estimated gradient $\widehat{\mu}_{z_t} \in \mathbb{R}^d$ satisfies $\|\widehat{\mu}_{z_t} - \mu_{z_t}\|_2 \le c\, \|z_t - x\|_2$ for $c = 4$. Then, we have $\|z_{t+1} - x\|_2^2 \le 0.99\, \|z_t - x\|_2^2$.*

*Proof Sketch of Lemma 5.3.* We provide a proof sketch and defer the full proof to Appendix B. Our objective function is nonconvex (even with infinitely many samples and no corruption). However, when the starting point $z_1$ is close to a global optimum, it is well-known that gradient descent is

well-behaved. More specifically, for any $z$ close to the ground truth $x$, we can show that the (expected) true gradient $\mu_z$ aligns with the direction toward $x$:

$$\langle \mu_z, z - x \rangle \geq 7.5 \, \|z - x\|_2^2 \quad \text{and} \quad \|\mu_z\|_2 \leq 29 \, \|z - x\|_2 \ ,$$

which is sufficient for proving geometric convergence. We can immediately see that this argument is robust to additive error in $\mu_z$ that is proportional to $\|z - x\|_2$. When $\|\widehat{\mu}_z - \mu_z\|_2 \leq c \, \|z - x\|_2$,

$$\langle \widehat{\mu}_z, z - x \rangle \geq (7.5 - c) \, \|z_t - x\|_2^2 \quad \text{and} \quad \|\widehat{\mu}_{z_t}\|_2 \leq (29 + c) \, \|z_t - x\|_2 \ .$$

When $c < 7.5$, we can choose an appropriate step size $\eta$ such that the distance between $z_t$ and $x$ decreases by a constant factor in each iteration. $\qquad\square$

We are now ready to prove Lemma 3.3, which states the performance guarantee, sample complexity, runtime, and success probability of Algorithm 2. We restate Lemma 3.3 before proving it.

**Lemma 3.3** (Robust Gradient Descent). *Consider the setting of Problem 1.3. Let $\Delta > 0$ be the desired precision. Let $0 < \epsilon < \epsilon_0$ for some sufficiently small universal constant $\epsilon_0$. Given an $\epsilon$-corrupted set of $n = \widetilde{\Omega}(d \log^2(1/\Delta))$ samples and an initial guess $z_1$ such that $r(z_1) = \min(\|z_1 - x\|_2, \|z_1 + x\|_2) \leq 1/8$, we can compute a vector $z \in \mathbb{R}^d$ in time $\widetilde{O}(nd)$ such that $r(z) \leq \Delta$ with probability at least $0.95$.*

*Proof of Lemma 3.3.* First, we prove the success probability of Algorithm 2. Algorithm 2 can fail in two ways: *(i)* if some $N_t$ has more than $(2\epsilon)$-fraction of corrupted samples, or *(ii)* if Lemma 5.2 fails in some iteration $t$. The probability of event *(i)* is at most $0.01$ for our choice of $n$, which follows from a standard application of Hoeffding's inequality and a union bound over $T$ iterations. For event *(ii)*, we choose a sufficiently small $\delta = O(1/T)$ in Lemma 5.2, so each robust gradient estimation fails with probability at most $O(\delta) = 0.01/T$, and overall the probability of event *(ii)* is at most $0.01$. For the rest of the proof, we assume these bad events do not happen.

Next, we show the correctness of Algorithm 2. Without loss of generality, we can assume that $z_1$ is closer to the ground truth $x$ than to $-x$, which implies $\|z_1 - x\|_2 \leq 1/8$. By Lemma 5.2, we can obtain an approximation $\widehat{\mu}_{z_1}$ of the true gradient $\mu_{z_1}$ at $z_1$ such that $\|\widehat{\mu}_{z_1} - \mu_{z_1}\|_2 \leq c \, \|z_1 - x\|_2$. Then by Lemma 5.3, we know that $\|z_2 - x\|_2 \leq 0.99 \, \|z_1 - x\|_2$ after one step of gradient descent. We can iteratively apply these two lemmas to show that, after $T = O(\log(1/\Delta))$ iterations, we have $\|z_{T+1} - x\|_2 \leq \Delta$. One technical issue is that in iteration $t$, we need to use a fresh subset of samples $N_t$. By the principle of deferred decisions, we can view $(a_i, y_i)_{i \in N_t}$ as being generated (and corrupted) after $z_t$ is chosen, which forms a $(2\epsilon)$-corrupted set of gradients at $z_t$.

Finally, we analyze the sample complexity and runtime of Algorithm 2. Algorithm 2 requires in total $n = mT = \Omega(d \log d \log^2(1/\Delta))$ samples. A random partition can be computed in $O(n)$ time by shuffling the input. In each iteration, the $m$ gradients in $N_t$ can be computed using Equation (7) in time $O(md)$, and $z_t$ can be updated in time $O(d)$. By Lemma 5.2, the true gradient can be robustly estimated in time $\widetilde{O}(md \log T) = \widetilde{O}(md \log \log(1/\Delta))$. The overall runtime of the algorithm is $\widetilde{O}(n + (md \log \log(1/\Delta))T) = \widetilde{O}(nd \log \log(1/\Delta)) = \widetilde{O}(nd)$. $\qquad\square$

## 6 Conclusions and Future Directions

In this paper, our main conceptual contribution is to propose and study the outlier-robust phase retrieval problem, where a constant fraction of the input data is corrupted. Notably, we allow corruption in both the sampled frequencies $a_i \in \mathbb{R}^d$ and the intensity measurements $y_i \in \mathbb{R}$. Our main technical contribution is the design and analysis of a near-sample-optimal and nearly-linear-time algorithm that solves this problem with provably guarantees.

An immediate technical question is whether our sample complexity can be tightened by removing some $\log(1/\Delta)$ factors. One potential approach is to open robust mean estimation algorithms instead of using them in a black-box manner. One could examine the stability conditions that these algorithms require, and see if these stability conditions can be proved without partitioning the samples and using fresh samples in each iteration. More broadly, we believe our framework can be used to develop outlier-robust algorithms for other tractable nonconvex problems, by first finding an initial solution in a saddle-free region near a global optimum and then converging to this global optimum using robust gradient descent.

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

 # A  Omitted Proofs in Section 4

**Lemma A.1.** *[Lemma 4.2, Formal].   For any $\delta_0 > 0$, there exists constants $\epsilon_0, c > 0$ such that when $n > c \cdot d \log d$ and we are given a set of $n$ $\epsilon$-corrupted samples, where $0 \leq \epsilon \leq \epsilon_0$, then with probability at least $0.98$, it holds*

$$\forall w \in \Delta_{n,2\epsilon}, \left\| Y_{G,w} - (I + 2xx^\top) \right\|_2 \leq \delta_0 . \tag{8}$$

*Proof of Lemma A.1.* We recall the definition of $Y_{G,w} = \sum_{i \in G} w_i y_i a_i a_i^\top$. Let $\ell = \epsilon \cdot n$ and let $\{(a_{n+i}, y_{n+i})\}_{i=1}^{\ell}$ be the set of samples that were removed by the $\epsilon$-corruption adversary. Let $G' = G \cup \{n+1, \ldots, n+\ell\}$, $n' = n + \ell$, and $\epsilon' = \epsilon/(1+\epsilon)$. Note that without loss of generality, we can assume that $|G| = (1-\epsilon)n$ and $|G'| = (1-\epsilon')n' = n$.

We define a mapping $\sigma : \Delta_{n,2\epsilon} \rightarrow \Delta_{n',3\epsilon'}$ such that

$$\sigma(w)_i = \begin{cases} w_i & i \in [n] \\ 0 & \text{otherwise} \end{cases} . \tag{9}$$

In other words, all the weights are the same for the samples with index in the set $[n]$, and are equal to $0$ for the samples removed by the adversary. We can verify that $\sigma(w) \in \Delta_{n',3\epsilon'}$ for all $w \in \Delta_{n,2\epsilon}$ since $\sigma(w)_i \leq w_i \leq 1/(1-2\epsilon)n = 1/(1-3\epsilon')n'$ for all $i \in [n']$, and $\|\sigma(w)\|_1 = \|w\|_1 = 1$. Furthermore, we have $Y_{G,w} = Y_{G',\sigma(w)}$ for all $w \in \Delta_{n,2\epsilon}$. We denote with $w^* \in \Delta_{n',3\epsilon'}$ the desired uniform weighting of the samples with index in $G'$, i.e., $w_i^* = \frac{1}{(1-\epsilon')n'} \mathbb{1}_{i \in G'}$.

It suffices to show both $\left\| Y_{G',w^*} - (I + 2xx^\top) \right\|_2 \leq \delta_0/2$ and $\left\| Y_{G',\sigma(w)-w^*} \right\|_2 \leq \delta_0/2$.

By triangle inequality, for any $w \in \Delta_{n,2\epsilon}$, it holds

$$\left\| Y_{G',\sigma(w)} - (I + 2xx^\top) \right\|_2 \leq \left\| Y_{G',w^*} - (I + 2xx^\top) \right\|_2 + \left\| Y_{G',\sigma(w)-w^*} \right\|_2 ,$$

Thus, it suffices to show both $\left\| Y_{G',w^*} - (I + 2xx^\top) \right\|_2 \leq \delta_0/2$ and $\left\| Y_{G',\sigma(w)-w^*} \right\|_2 \leq \delta_0/2$.

We upper bound the first term. By using the definition of $w^*$, note that

$$\left\| Y_{G',\sigma(w)} - (I + 2xx^\top) \right\|_2 = \left\| \sum_{i \in G'} w_i^* y_i a_i a_i^\top - (I + 2xx^\top) \right\|_2$$

$$= \left\| \sum_{i \in G'} \frac{1}{|G'|} y_i a_i a_i^\top - (I + 2xx^\top) \right\|_2 .$$

Since $\mathbb{E}(y_i a_i a_i^\top) = I + 2xx^\top$ for any $i \in G'$, we can use a concentration inequality to upper bound this term. By Lemma A.2, as long as $n \geq c_1(\delta_0/2) \cdot d \log d$, with probability at least $0.995$, we have

$$\left\| Y_{G',w^*} - (I + 2xx^\top) \right\|_2 \leq \delta_0/2 . \tag{10}$$

It remains to show a high-probability upper bound to the second term $\left\| Y_{G',w^*-\sigma(w)} \right\|_2 \leq \delta_0/2$. The first observation is that the weighting $w^*$ and $\sigma(w)$ for any $w \in \Delta_{n,2\epsilon}$ cannot too different. In particular, we can show the following upper bound:

$$\sum_{i \in G'} |w_i^* - \sigma(w)_i| \leq \sum_{i=1}^{n'} |w_i^* - \sigma(w)_i| \leq \sup_{w,w' \in \Delta_{n',3\epsilon'}} \sum_{i=1}^{n'} |w_i - w_i'| \tag{11}$$

We observe that $\Delta_{n',3\epsilon'}$ can be seen as the convex combination of all possible uniform weighting over subsets of $n'(1-3\epsilon')$ samples. Thus, the maximum distance will be between two points of the convex hull, and we can upper bound (11) as:

$$\sum_{i \in G'} |w_i^* - \sigma(w)_i| \leq \sup_{w,w' \in \Delta_{n',3\epsilon'}} \sum_{i=1}^{n'} |w_i - w_i'| \leq \frac{6\epsilon' n}{n'(1-3\epsilon')} \leq 6\epsilon . \tag{12}$$

For a fixed unit vector $z \in \mathbb{S}^{d-1}$ with $z = px + qu$ where $u \in \mathbb{S}^{d-1}$ and $\langle u, x \rangle = 0$, we have:

$$\max_{w \in \Delta_{n,2\epsilon}} \left| z^\top Y_{G', w^* - \sigma(w)} z \right| = \max_w \left| \sum_{i \in G'} (w_i^* - \sigma(w)_i) \langle a_i, x \rangle^2 \langle a_i, z \rangle^2 \right|$$

$$= \max_w \left| \sum_{i \in G'} (w_i^* - \sigma(w)_i) \langle a_i, x \rangle^2 (p \langle a_i, x \rangle + q \langle a_i, u \rangle)^2 \right|$$

$$\leq 2 \max_w \left| \sum_{i \in G'} (w_i^* - \sigma(w)_i)(\langle a_i, x \rangle^4 + \langle a_i, x \rangle^2 \langle a_i, u \rangle^2) \right|$$

$$\leq 2 \max_w \sum_{i \in G'} |w_i^* - \sigma(w)_i| \langle a_i, x \rangle^4$$

$$+ 2 \max_w \sum_{i \in G'} |w_i^* - \sigma(w)_i| \langle a_i, x \rangle^2 \langle a_i, u \rangle^2 \quad .$$

For ease of notation, let $\beta_i \doteq |w_i^* - \sigma(w)_i|$. Observe that $0 \leq \beta_i \leq \frac{1}{(1-2\epsilon)n}$ for all $i$, and $\sum_i \beta_i \leq 6\epsilon$ due to (11). We have:

$$\max_{w \in \Delta_{n,2\epsilon}} \left| z^\top Y_{G', w^* - \sigma(w)} z \right| \leq 2 \max_{\beta: \sum_{i \in G'} \beta_i \leq 6\epsilon \text{ and } 0 \leq \beta_i \leq \frac{1}{(1-2\epsilon)n}} \sum_{i \in G'} \beta_i \langle a_i, x \rangle^4$$

$$+ 2 \max_{\beta: \sum_{i \in G'} \beta_i \leq 6\epsilon \text{ and } 0 \leq \beta_i \leq \frac{1}{(1-2\epsilon)n}} \sum_{i \in G'} \beta_i \langle a_i, x \rangle^2 \langle a_i, u \rangle^2$$

$$\leq \frac{2}{(1 - 2\epsilon)n} \max_{L \subseteq G', |L| = 6\epsilon n} \sum_{i \in L} \langle a_i, x \rangle^4$$

$$+ \frac{2}{(1 - 2\epsilon)n} \max_{L \subseteq G', |L| = 6\epsilon n} \sum_{i \in L} \langle a_i, x \rangle^2 \langle a_i, u \rangle^2 . \tag{13}$$

Inequality (13) follows by assigning the maximum possible $\beta_i$ to the largest entries of the sum until we hit the budget $6\epsilon$ due to (11).

We bound $\max_L \sum_{i \in L} \langle a_i, x \rangle^4$ first. Let $X_i = \langle a_i, x \rangle \sim \mathcal{N}(0, 1)$ for $i \in G'$, and define the threshold function $h_r(z) = \begin{cases} 0, & z \leq r \\ z, & z > r \end{cases}$ for $r = C^2 \cdot \ln^2(1/\epsilon)$ with constant $C > 0$ to be determined.

Note that $z \leq r + h_r(z)$ for all $z > 0$. Therefore,

$$\max_{L \subseteq G', |L| = 6\epsilon n} \frac{1}{n} \sum_{i \in L} X_i^4 \leq \max_L \frac{1}{n} \sum_{i \in L} r + \max_L \frac{1}{n} \sum_{i \in L} h_r(X_i^4)$$

$$\leq 6\epsilon r + \frac{1}{n} \sum_{i \in G'} h_r(X_i^4).$$

Then, we consider to bound $\exp\left( \sum_{i \in G'} c \cdot h_r(X_i^4) \right)$ for some $c > 0$ to be determined. For any $i \in G'$ and $z \geq 1$, with $C = 6$, for all $\epsilon > 0$, we have

$$\mathbf{Pr}\left[ \exp\left(c \cdot h_r(X_i^4)\right) \geq z \right] \leq \mathbf{Pr}\left[ h_r(X_i^4) \geq 0 \right] = \mathbf{Pr}\left[ X_i \geq r^{1/4} \right] \leq \exp(-\sqrt{r}/2)$$

$$\leq \exp(\ln \epsilon \cdot C/2)$$

$$\leq \epsilon^3.$$

At the same time, with $c < 1/200$, for all $z \geq 1$, we have

$$\mathbf{Pr}\left[ \exp\left(c \cdot h_r(X_i^4)\right) \geq z \right] \leq \mathbf{Pr}\left[ \exp(c \cdot X_i^4) \geq z \right] \leq \mathbf{Pr}\left[ X_i^4 \geq \frac{\ln z}{c} \right]$$

$$\leq \exp\left( -\sqrt{\frac{\ln z}{c}}/2 \right)$$

$$\leq z^{-3}.$$

508      Therefore, $\mathbf{Pr}\left[\exp\left(c \cdot h_r(X_i^4)\right) \geq z\right] \leq \min\{z^{-3}, \epsilon^3\}$, and for all $\epsilon < 1/2$, we have

$$
\begin{aligned}
\mathbf{E}\left[\exp\left(c \cdot h_r(X_i^4)\right)\right] &= \int_0^\infty \mathbf{Pr}\left[\exp\left(c \cdot h_r(X_i^4)\right) \geq z\right] \mathrm{d}z \\
&\leq \int_0^1 1 \mathrm{d}z + \int_1^{1/\epsilon} \epsilon^3 \mathrm{d}z + \int_{1/\epsilon}^\infty z^{-3} \mathrm{d}z \\
&= 1 + \epsilon^2 - \epsilon^3 + \frac{1}{2}\epsilon^2 \\
&\leq 1 + \epsilon^2 \\
&\leq \exp(\epsilon^2).
\end{aligned}
$$

509      Since $\{X_i\}_{i\in G'}$ are independent, we have

$$
\mathbf{E}\left[\exp\left(\sum_{i\in G'} c \cdot h_r(X_i^4)\right)\right] = \mathbf{E}\left[\prod_{i\in G'} \exp\left(c \cdot h_r(X_i^4)\right)\right] \leq \exp(\epsilon^2 n).
$$

510      Finally, by Markov's inequality, for any constant $\delta_1 > 0$, as long as $\epsilon \leq \sqrt{\delta_1 c}$, we have

$$
\begin{aligned}
\mathbf{Pr}\left[\sum_{i\in G'} h_r(X_i^4) \geq 2\delta_1 n\right] &= \mathbf{Pr}\left[\exp\left(\sum_{i\in G'} c \cdot h_r(X_i^4)\right) \geq \exp(2\delta_1 cn)\right] \\
&\leq \exp(\epsilon^2 n - 2\delta_1 cn) \leq \exp(-\delta_1 cn).
\end{aligned}
$$

511      As a result, with sufficiently large $n \geq c_2(\delta_1) \cdot d \log d$ and sufficiently small $\epsilon$ such that $6\epsilon r \leq \delta_1$,

$$
\mathbf{Pr}\left[\epsilon r + \frac{1}{n}\sum_{i\in G'} h_r(X_i^4) \geq 3\delta_1\right] \leq \exp(-\delta_1 cn) \leq 0.995 \cdot 9^{-d}.
$$

512      Also, $\max_L \sum_{i\in L} \langle a_i, x\rangle^2 \langle a_i, u\rangle^2$ has a similar tail bound and therefore can be bounded in the same
513      way. We can then bound the operator norm via an epsilon-net argument. Set $\delta_1 = \delta_0/24$. By an
514      $1/4$-net on $\mathcal{S}^{d-1}$ with $|\mathcal{N}| \leq 9^d$, we have that

$$
\mathbf{Pr}\left[\max_{z\in\mathbb{S}^{d-1}} \left|z^\top Y_{G',w^*-\sigma(w)} z\right| \geq \delta_0/2\right] \leq 9^d \cdot 0.99 \cdot 9^{-d} \leq 0.99.
$$

515      By combining the above inequality with Equation (10), we can conclude that, for any $\delta_0 > 0$, there
516      exists $\epsilon_0 > 0$, such that when $n \geq \max\{c_1(\delta_0), c_2(\delta_0)\} \cdot d \log d$ and $0 \leq \epsilon \leq \epsilon_0$, with probability at
517      least $0.98$,

$$
\forall w \in \Delta_{n,2\epsilon}, \left\|Y_{G,w} - (I + 2xx^\top)\right\|_2 = \left\|Y_{G',\sigma(w)} - (I + 2xx^\top)\right\|_2 \leq \delta_0.
$$

518      Therefore, with probability at least $0.98$, we have $\left\|Y_{G,w} - (I + 2xx^\top)\right\|_2 \leq \delta_0$ for all $w \in \Delta_{n,2\epsilon}$.
519      $\square$

## A.1    Concentration Inequalities

521      For the undisturbed samples, we have the following concentration result.

522      **Lemma A.2** ([3] Section A.4.2). *Let $x \in \mathbb{R}^d$. For any $\delta > 0$, there exists a constant $C(\delta) > 0$*
523      *such that when $n > C \cdot d \log d$ and we are given a set of $n$ samples $\{(a_i, y_i)\}_{i=1}^n$ with $a_i \sim \mathcal{N}(0, I)$*
524      *independently and $y_i = \langle a_i, x\rangle^2$ for all $i \in [n]$, then with probability at least $0.99$, it holds*

$$
\left\|\frac{1}{n}\sum_{i=1}^n y_i a_i a_i^\top - (I + 2xx^\top)\right\|_2 \leq \delta.
$$

## B  Omitted Proofs in Section 5

**Lemma 5.1.** *Let $\mathcal{D}_z$ be the distribution of gradients at $z$ as defined in Equation* (7). *For any $z \in \mathbb{R}^d$ with $\|z - x\|_2 \leq 1$, the covariance matrix $\Sigma_z$ of $\mathcal{D}_z$ satisfies $\Sigma_z \preceq O(\|z - x\|_2^2)I$.*

*Proof of Lemma 5.1.* Recall that $g \sim \mathcal{D}_z$ is distributed as

$$g = \frac{\partial}{\partial z}\left[\left(\langle a, z\rangle^2 - \langle a, x\rangle^2\right)^2\right]$$
$$= -4\left(\langle a, z\rangle^2 - \langle a, x\rangle^2\right)\langle a, z\rangle a \quad \text{where } a \sim \mathcal{N}(0, 1) .$$

Because $\mathbf{E}_{g \sim \mathcal{D}_z}[g] = \mu_z$, the spectral norm of $\Sigma_z$ can be upper bounded as follows:

$$\|\Sigma_z\|_2 = \left\|\mathop{\mathbf{E}}_{g \sim \mathcal{D}_z}\left[gg^\top\right] - \mu_z \mu_z^\top\right\|_2 \leq \left\|\mathop{\mathbf{E}}_{g \sim \mathcal{D}_z}\left[gg^\top\right]\right\|_2 .$$

Consequently, it suffices to upper bound $\left\|\mathbf{E}_{g \sim \mathcal{D}_z}\left[gg^\top\right]\right\|_2$. Let $h = z - x$.

$$\left\|\mathop{\mathbf{E}}_{g \sim \mathcal{D}_z}\left[gg^\top\right]\right\|_2 = \max_{\|v\|_2 = 1} v^\top \mathop{\mathbf{E}}_{g \sim \mathcal{D}_z}\left[gg^\top\right] v = \max_{\|v\|_2 = 1} \mathop{\mathbf{E}}_{g}\left[\langle g, v\rangle^2\right]$$
$$= 16 \max_{\|v\|_2 = 1} \mathop{\mathbf{E}}_{a \sim \mathcal{N}(0, I)}\left[(\langle a, z\rangle^2 - \langle a, x\rangle^2)^2 \langle a, z\rangle^2 \langle a, v\rangle^2\right]$$
$$= 16 \max_{\|v\|_2 = 1} \mathop{\mathbf{E}}_{a \sim \mathcal{N}(0, I)}\left[\langle a, h\rangle^2 \langle a, 2x + h\rangle^2 \langle a, x + h\rangle^2 \langle a, v\rangle^2\right]$$
$$\leq 16 \max_{\|v\|_2 = 1}\left(\mathop{\mathbf{E}}_{a}\left[\langle a, h\rangle^8\right] \mathop{\mathbf{E}}_{a}\left[\langle a, 2x + h\rangle^8\right] \mathop{\mathbf{E}}_{a}\left[\langle a, x + h\rangle^8\right] \mathop{\mathbf{E}}_{a}\left[\langle a, v\rangle^8\right]\right)^{1/4}$$
$$= 16 \max_{\|v\|_2 = 1}\left(105^4 \|h\|_2^8 \|2x + h\|_2^8 \|x + h\|_2^8 \|v\|_2^8\right)^{1/4}$$
$$= (16 \cdot 105) \|h\|_2^2 \|2x + h\|_2^2 \|x + h\|_2^2 = O(\|h\|_2^2) .$$

The first inequality is due to Cauchy-Schwarz inequality. The last step uses the fact that $\|x\|_2 = 1$ and $\|h\|_2 \leq 1$. $\qquad\square$

**Lemma 5.3.** *Suppose in iteration $t$ of Algorithm 2, the current solution $z_t$ satisfies $\|z_t - x\|_2 \leq 1/8$, and the estimated gradient $\widehat{\mu}_{z_t} \in \mathbb{R}^d$ satisfies $\|\widehat{\mu}_{z_t} - \mu_{z_t}\|_2 \leq c\|z_t - x\|_2$ for $c = 4$. Then, we have $\|z_{t+1} - x\|_2^2 \leq 0.99\|z_t - x\|_2^2$.*

*Proof of Lemma 5.3.* Recall that $g \sim \mathcal{D}_z$ is distributed as

$$g = \frac{\partial}{\partial z}\left[\left(\langle a, z\rangle^2 - \langle a, x\rangle^2\right)^2\right] \quad \text{where } a \sim \mathcal{N}(0, I) .$$

We can compute the mean of $\mathcal{D}_z$ using moments of Gaussian:

$$\mu_z = \mathop{\mathbb{E}}_{g \sim \mathcal{D}_z} g = \left(12\|z\|_2^2 - 4\|x\|_2^2\right)z - 8\langle x, z\rangle x . \tag{14}$$

Consider one step of gradient descent in Algorithm 2: $z_{t+1} = z_t - \eta\widehat{\mu}_{z_t}$, where $\widehat{\mu}_{z_t}$ is an approximate gradient. We have

$$\|z_{t+1} - x\|_2^2 = \|z_t - \eta\widehat{\mu}_{z_t} - x\|_2^2 = \|z_t - x\|_2^2 - 2\eta\langle\widehat{\mu}_{z_t}, z_t - x\rangle + \eta^2\langle\widehat{\mu}_{z_t}, \widehat{\mu}_{z_t}\rangle$$

To prove convergence, we need to lower bound $\langle\widehat{\mu}_{z_t}, z_t - x\rangle$ and upper bound $\langle\widehat{\mu}_{z_t}, \widehat{\mu}_{z_t}\rangle$.

We write $z = z_t$ and $h = z - x$ to simplify notation. We can substitute $z = x + h$ in Equation (14):

$$\mu_z = \left(16\langle x, h\rangle + 12\|h\|_2^2\right)x + \left(8\|x\|_2^2 + 24\langle x, h\rangle + 12\|h\|_2^2\right)h .$$

542   Recall the assumptions of this lemma: $\|x\|_2 = 1$, $\|h\|_2 \leq 1/8$, and $\|\widehat{\mu}_z - \mu_z\|_2 \leq c \|h\|_2$.

543   First we lower bound $\langle \widehat{\mu}_z, h \rangle$.

$$
\begin{aligned}
\langle \widehat{\mu}_z, h \rangle &= \langle \mu_z, h \rangle + \langle \widehat{\mu}_z - \mu_z, h \rangle \\
&= 16 \langle x, h \rangle^2 + 36 \langle x, h \rangle \|h\|_2^2 + 8 \|x\|_2^2 \|h\|_2^2 + 12 \|h\|_2^4 + \langle \widehat{\mu}_z - \mu_z, h \rangle \\
&\geq -\tfrac{81}{4} \|h\|_2^4 + 8 \|x\|_2^2 \|h\|_2^2 + 12 \|h\|_2^4 - \langle \widehat{\mu}_z - \mu_z, h \rangle \\
&\geq \left(8 - \tfrac{33}{256} - c\right) \|h\|_2^2 \\
&\geq (7.5 - c) \|h\|_2^2 \ .
\end{aligned}
$$

544   The first inequality uses the fact that $16 \langle x, h \rangle^2 + 36 \langle x, h \rangle \|h\|_2^2$ is a second-order polynomial of
545   $\langle x, h \rangle$, which has minimum value $-\tfrac{81}{4} \|h\|_2^4$ for all $\langle x, h \rangle \in \mathbb{R}$.

546   Next we upper bound $\langle \widehat{\mu}_z, \widehat{\mu}_z \rangle$ using the triangle inequality.

$$
\begin{aligned}
\|\widehat{\mu}_z\|_2 &\leq \|\mu_z\|_2 + \|\widehat{\mu}_z - \mu_z\|_2 \\
&\leq \left(16 \langle x, h \rangle + 12 \|h\|_2^2\right) \|x\|_2 + \left(8 \|x\|_2^2 + 24 \langle x, h \rangle + 12 \|h\|_2^2\right) \|h\|_2 + c \|h\|_2 \\
&\leq \left(16 + \tfrac{12}{8} + 8 + \tfrac{24}{8} + \tfrac{12}{64} + c\right) \|h\|_2 \\
&\leq (29 + c) \|h\|_2 \ .
\end{aligned}
$$

547   Putting everything together, we have

$$
\begin{aligned}
\|z_{t+1} - x\|_2^2 &= \|z_t - x\|_2^2 - 2\eta \langle \widehat{\mu}_{z_t}, z_t - x \rangle + \eta^2 \langle \widehat{\mu}_{z_t}, \widehat{\mu}_{z_t} \rangle \\
&\leq \left[1 - 2(7.5 - c)\eta + (29 + c)^2 \eta^2\right] \|z_t - x\|_2^2 \ .
\end{aligned}
$$

548   Choosing $c = 4$ and $\eta = 1/300$ gives that $\|z_{t+1} - x\|_2^2 \leq 0.99 \|z_t - x\|_2^2$.    $\square$

## 549  C  Counter-examples

550   Prior robust phase retrieval algorithms [24, 45] focus on the setting where the observations $y_i$'s are sub-
551   ject to adversarial perturbation while the measuring vectors $a_i$'s are independently sampled from the
552   Gaussian distribution. The Median Truncated Wirtinger Flow Algorithm [45] first initialize $z^{(0)}$ by the
553   spectral method, calculating $z^{(0)}$ as the top eigenvector of $Y := \frac{1}{m} \sum_{i=1}^m y_i a_i a_i^\top \mathbb{1}_{|y_i| \leq \alpha^2 \, \mathrm{med}(\{y_i\}_{i=1}^m)}$
554   using a truncated set of samples, where the threshold is determined by $\mathrm{med}(\{y_i\}_{i=1}^m)$, the median
555   over all $y_i$'s. As long as the fraction of of outliers is not too large and the sample complexity is large
556   enough, the initialization is guaranteed to be within a small neighborhood of the ground truth.

557   In this section, we present a counter-example where robust phase retrieval algorithms [24, 45] can be
558   insufficient when directly applied to the $\epsilon$-corruption phase retrieval problem.

559   Let $x \in \mathbb{S}^{d-1}$ be the ground truth unit vector. Here we construct an $\epsilon$-corruption adversary that can
560   manipulate the top eigenvector of the empirical covariance matrix $Y = \sum_{i=1}^n y_i a_i a_i^\top$, even when $y_i$
561   are accurately calculated as $y_i = (a_i^\top x)^2$.

562   Let $u \in \mathbb{S}^{d-1}$ be a unit vector such that $x^\top u = 0$. Suppose the adversary changes 1% of the $a_i$'s
563   to $a_i = \sqrt{d - 1/25} \cdot u + 1/5x$, and suppose all the $y_i$'s are accurate. In particular, the length of
564   the corrupted $a_i$'s is comparable to Gaussian vectors, and the corresponding $y_i = (a_i^\top x)^2 = 1/25$.
565   Consequently, the median-truncated initialization in [45] will not be able to filter out such $y_i$. However,
566   the top eigenvector of $\mathbf{E}[Y] = \mathbf{E}\left[\sum_{i=1}^n y_i a_i a_i^\top\right] = O(d) u u^\top + O(\sqrt{d})(u x^\top + x u^\top) + O(1)(I +$
567   $2xx^\top)$ will be manipulated to $u$, which is far from the ground truth $x$.

