# OpenReview forum: "Outlier-Robust Phase Retrieval in Nearly-Linear Time"
_NeurIPS.cc/2024/Conference — Submitted to NeurIPS 2024_

### Official Review · Reviewer_ednt · 2024-07-08

**Soundness:** 2
**Presentation:** 3
**Contribution:** 2
**Rating:** 2
**Confidence:** 5

**Summary:**

This paper addresses the challenge of achieving outlier robustness in phase retrieval, specifically focusing on the recovery of real-valued signals from intensity measurements that have been corrupted by adversarial outliers. The contribution of this work is the development of a nearly-linear time algorithm that is nearly sample-optimal and can accurately recover the true vector despite the presence of outliers. This is done through a two-step process that involves robust spectral initialization and robust gradient descent, utilizing recent results in high-dimensional robust statistics.

**Strengths:**

+A two-stage algorithm achieves nearly-linear time complexity consisting of an initial spectral initialization phase and a gradient descent refinement phase.

+Theoretical analysis showing the algorithm can recover the ground truth signal despite the presence of outliers, while maintaining near-optimal sample efficiency.

**Weaknesses:**

- The paper's theoretical results assume the corruption level $\epsilon$ as a constant in Theorem 3.1, despite it being initially introduced as a variable in Definition 1.2 to denote the extent of sample corruption. This constant treatment affects the robustness of the results by neglecting the influence of $\epsilon$ on the sample complexity. A thorough analysis that explicitly considers the variability of $\epsilon$ would significantly strengthen the theoretical foundation.
- Lack of numerical validations for the theoretical claims
- The idea and design of the two-stage robust phase retrieval algorithm is not novel, by adapting existing reweighting phase retrieval algorithms (throught different design of the reweights) to handle outliers through robust statistics. The contamination model is also from existing works, which has been recent studied in a number of different contexts; e.g., robust linear/nonlinear estimation by e.g., Diakonikolas and coauthors.
- Lack of numerical validation and comparison of the proposed algorithm with respect to existing (robust) Gaussian phase retrieval algorithms. It is difficult to judge if the proposed algorithm is of practical interest (or only of theoretical interest).
- Quite a lot statements in the paper are rather confusing, e.g., "we propose the problem of outlier robust phase retrieval"; robust phase retrieval has been long studied in the literature; as far as I understand, the paper studies a new robust phase retrieval problem by considering also adversarial a_i's on top of existing formualtions. ii) "It is well-known that natural nonconvex formulations of phase retrieval do not have spurious local optima." which is not precise enough and is true under very stringent assumptions. iii) "This is first achieved via approaches based on semidefinite programming (SDP) relaxations (see, e.g., Cand`es et al. (2015c))." I guess the first Gaussian phase retrieval algorithm was the AltMin algorithm (Netrapalli, et al 2013. Phase retrieval using alternating minimization. Advances in Neural Information Processing Systems, 26.) which was interestingly not cited in the submission. iv) "Similar landscape results are known for other natural nonconvex formulations of phase retrieval as well (e.g., min f(z) = P i( √ yi − |⟨ai, z⟩|)2 (Soltanolkotabi, 2019))." The first work to study the Gaussian phase retrieval based on the magnitude-based least-squares nonconvex formulation and achieve provable guarantees is (Wang et al (2017). Solving systems of random quadratic equations via truncated amplitude flow. IEEE Transactions on Information Theory. 64(2):773-94.) Please be careful and fair in stating related results.

**Questions:**

i) The main motivation of the paper is to study a new robust phase retrieval (RPR) problem where in addition to the existing well-studied RPR (earlier in e.g., Hand, et al. 2016. Corruption robust phase retrieval via linear programming. arXiv preprint arXiv:1612.03547.), the sampling vectors a_i's can also be adversarially manipuated. Nonetheless, since PR is quite "simple" model, it can be imagined some of the effect of the adversarial a_i's can also be lumped into the adversarial y_i's. I am not sure whether the problem here is indeed more challenging than existing corruption pr considering only adversarial y_i's. If not rigorously theoretically analyzed, numerical comparisons with these existing rpr algorithms will be needed and appreciated.
ii) The designed algorithm is a two-stage reweighting algorithm that consists of a reweighting spectral initialization and reweighting gradient descent algorithm; nonetheless, this unique design and use of reweighting to improve nonconvex PR performance has been initially documented in [Yuan, et al (2017). Phase retrieval via reweighted Wirtinger flow. Applied Optics, 56(9), 2418-2427; Wang, et al (2018). Phase retrieval via reweighted amplitude flow. IEEE Transactions on Signal Processing. 66(11):2818-33.]; the form of using the weights here is exactly the same as in the two papers]; The only difference is the specific option/design of the values of the weights, certainly for considerations of handling the outliers here. It is very interesting to understand what is the original motivation of using and where does the reweighting technique come from.
iii) Last but not the least, from a practical perspective, say e.g., the Fourier/modulated phase retrieval problem using random masks, it makes sense that possiblem (adversarial) noise may be present in measuring the intensity measurements; but it is not clear why the sampling vectors that correspond to rows of DFT matrices (or multiplied by random masking +1 or -1 values) could also be manupulated experimentally. Please provide a more practical setup validating the motivation of studying such a RPR problem.
iv) A lot of typos or inaccurate statements. e.g., "we propose and study..."; the claim "into a saddle-free region" is not theoretically and rigorously established.

---

> ### Author Rebuttal · Authors · 2024-08-07
>
> We would like to clarify that many of the reviewer's comments have been addressed in our revisions. Several "quotes" provided by the reviewer no longer appear in our current draft.
>
> ---
>
> **R**: The paper's theoretical results assume the corruption level $\epsilon$ as a constant in Theorem 3.1, despite it being initially introduced as a variable in Definition 1.2 to denote the extent of sample corruption. This constant treatment affects the robustness of the results by neglecting the influence of $\epsilon$ on the sample complexity. A thorough analysis that explicitly considers the variability of $\epsilon$ would significantly strengthen the theoretical foundation.
>
> **A**: The reviewer's claim is false. In our work, we do not assume that the corruption level $\epsilon$ is constant and we do not neglect $\epsilon$ in the sample complexity. Our results are true as stated: they hold for any corruption level $\epsilon < \epsilon'$.
>
> ---
>
> **R**: The idea and design of the two-stage robust phase retrieval algorithm is not novel, by adapting existing reweighting phase retrieval algorithms (throught different design of the reweights) to handle outliers through robust statistics. The contamination model is also from existing works, which has been recent studied in a number of different contexts; e.g., robust linear/nonlinear estimation by e.g., Diakonikolas and coauthors.
>
> **A**: We do not claim that the contamination model is our contribution. Our work follows a long line of research where it is shown how to solve important machine learning problems robustly within this contamination model.  To our knowledge, our work is the first to solve study the phase retrieval problem in this model, and moreover, our algorithm simultaneously achieves near-optimal error, sample complexity, and runtime.
>
> ---
>
> **R**: The main motivation of the paper is to study a new robust phase retrieval (RPR) problem where in addition to the existing well-studied RPR (earlier in e.g., Hand, et al. 2016. Corruption robust phase retrieval via linear programming. arXiv preprint arXiv:1612.03547.), the sampling vectors a_i's can also be adversarially manipuated. Nonetheless, since PR is quite "simple" model, it can be imagined some of the effect of the adversarial a_i's can also be lumped into the adversarial y_i's. I am not sure whether the problem here is indeed more challenging than existing corruption pr considering only adversarial y_i's.
>
> **A**: The “RPR” work referenced by the reviewer solves a different and simpler problem where the corruption is only allowed in the sampling vector $a_i$, where in our work we allow arbitrary $\epsilon$-corruption in the input pairs $(a_i, y_i)$.
>
> Our setting is more challenging. In appendix C, we gave a counter-example to show why algorithms for “RPR”  [1][2] fail in our setting.
>
> ---
>
> **R**: The designed algorithm is a two-stage reweighting algorithm that consists of a reweighting spectral initialization and reweighting gradient descent algorithm; nonetheless, this unique design and use of reweighting to improve nonconvex PR performance has been initially documented [...]
>
> **A**: We stated in Line 91 that our two-stage algorithm is inspired by previous work. However, there is no corruption in these prior works. It is unclear how those references undermine the novelty of our contribution: The goal of re-weighting in the first stage of our algorithm is to obtain a “good” initial solution despite corruption. This is completely different from works referenced by the reviewer, which does not handle corruption.
>
> ---
>
> **R**: Quite a lot statements in the paper are rather confusing, e.g., "we propose the problem of outlier robust phase retrieval"; robust phase retrieval has been long studied in the literature; as far as I understand, the paper studies a new robust phase retrieval problem by considering also adversarial a_i's on top of existing formualtions.
>
> **A**: No previous work considered or studied the problem that we defined in our work where corruption is allowed in both $(a_i,y_i)$. We agree with the reviewer that "the paper studies a new robust phase retrieval problem". We name it "outlier robust phase retrieval" and we are justified to claim that we propose and study this problem because it has not been studied before.
>
> ---
>
> **R**: Other statements confusing
>
>  ii) "It is well-known that natural nonconvex formulations of phase retrieval do not have spurious local optima." which is not precise enough and is true under very stringent assumptions.
>
> iii) "This is first achieved via approaches based on semidefinite programming (SDP) relaxations (see, e.g., Cand`es et al. (2015c))." I guess the first Gaussian phase retrieval algorithm was the AltMin algorithm (Netrapalli, et al 2013. Phase retrieval using alternating minimization. Advances in Neural Information Processing Systems, 26.) which was interestingly not cited in the submission.
>
>  iv) "Similar landscape results are known for other natural nonconvex formulations of phase retrieval as well (e.g., min f(z) = P i( √ yi − |⟨ai, z⟩|)2 (Soltanolkotabi, 2019))." The first work to study the Gaussian phase retrieval based on the magnitude-based least-squares nonconvex formulation and achieve provable guarantees is [...] Please be careful and fair in stating related results.
>
> **A**: None of these "quotes" appear in the current draft.

---

### Official Review · Reviewer_BQYS · 2024-07-11

**Soundness:** 2
**Presentation:** 2
**Contribution:** 2
**Rating:** 4
**Confidence:** 4

**Summary:**

This paper focuses on the problem of outlier robust phase retrieval, whose goal is to recover a vector $x \in \mathbb{R}^d$ from $n$ intensity measurements $y_i = (a_i^\top x)^2$ when a small fraction of the samples are adversarially corrupted. The authors propose and study this problem, providing a nearly sample-optimal and nearly linear-time algorithm to recover the ground-truth vector $x$ in the presence of outliers.

**Strengths:**

This paper provides an analysis of the practically interesting problem of outlier robust phase retrieval. The algorithm and framework might have implications for various applications that are relevant to phase retrieval.

**Weaknesses:**

1. The analysis appears to be more incremental in nature compared to those in previous theoretical works related to phase retrieval. More precisely, the key novelty of the spectral initialization step resides in the assignment of a nonnegative weight to each sample. Regarding the gradient descent step, the problem seems to be simplified to the analysis of robust mean estimation algorithms.

2. Important references are missing. For instance, the authors ought to cite the works related to robust compressed sensing (and it would be better to discuss in more detail the disparities between the analysis for the robust gradient descent step in this work and the analysis in these relevant works), such as

- Liu, Liu, Yanyao Shen, Tianyang Li, and Constantine Caramanis. "High dimensional robust sparse regression." In International Conference on Artificial Intelligence and Statistics, pp. 411-421. PMLR, 2020.
- Liu, Liu, Tianyang Li, and Constantine Caramanis. "High Dimensional Robust $ M $-Estimation: Arbitrary Corruption and Heavy Tails." arXiv preprint arXiv:1901.08237 (2019).

3. The authors solely consider the scenario of noiseless intensity measurements and fail to take into account the noisy case.

4. The paper does not include experimental results, which could limit the confidence in the practical effectiveness of the proposed approach.

**Questions:**

Why in Theorem 1.4 (or its formal version in Theorem 3.1), there is no dependence on $\epsilon$ in both the sample complexity and the upper bound of $\min (\\|z-x\\|_2,\\|z+x\\|_2)$?

**Limitations:**

No experimental validations.

---

> ### Author Rebuttal · Authors · 2024-08-06
>
> We thank the reviewer for their comments and feedback.
>
> ---
>
> **R**: Why in Theorem 1.4 (or its formal version in Theorem 3.1), there is no dependence on $\epsilon$  in both the sample complexity and the upper bound on the error?
>
> **A**: Our proofs are formally correct and show that there is no dependence on $\epsilon$ in our sample complexity and error guarantee. This is the main contribution of our work. Without corruption, $O(d)$ samples are sufficient for phase retrieval, and we show how to achieve exact recovery (in nearly-linear time) using $\tilde O(d)$ samples even if a small $\epsilon$ fraction of input pairs $(a_i,y_i)$ are corrupted.
>
> Intuitively, this is information-theoretically possible because there are indeed enough “clean” samples. Algorithmically, this is possible because as we get closer to the true vector, the covariance matrix of the true gradients becomes smaller (formally stated as Lemma 5.1), which allows us to obtain increasingly precise gradient estimates, which enables recovery to arbitrary precision.
>
> ---
>
> **R**: The analysis appears to be more incremental in nature compared to those in previous theoretical works related to phase retrieval. More precisely, the key novelty of the spectral initialization step resides in the assignment of a nonnegative weight to each sample. Regarding the gradient descent step, the problem seems to be simplified to the analysis of robust mean estimation algorithms.
>
> **A**: Our algorithm achieves near-optimal error, sample complexity, and running time. Achieving near-optimality in all three aspects simultaneously is often a challenging task. The fact that it is not immediately obvious to the readers why the error and sample complexity do not depend on $\epsilon$ further suggests that our results are not straightforward.
>
> Moreover, our black-box use of robust mean estimation algorithms can be a plus, because it offers a simple and novel framework for solving tractable non-convex problems in the $\epsilon$-corruption model: instead of designing global algorithms (i.e., without initialization) that are robust, it might be easier to use a robust initialization step followed by black-box robust gradient descent.
>
> ---
>
> **R**: Important references are missing. For instance, the authors ought to cite the works related to robust compressed sensing (and it would be better to discuss in more detail the disparities between the analysis for the robust gradient descent step in this work and the analysis in these relevant works),
>
> **A**: We thank the reviewers for providing these references and we will add discussions on this. We note that we cited the following papers, which are often mentioned as the first to consider the idea of robust gradient descent in high-dimensional robust statistics literature:
>
> [1] Diakonikolas, I., Kamath, G., Kane, D., Li, J., Steinhardt, J., & Stewart, A.. SEVER: A Robust Meta-Algorithm for Stochastic Optimization. ICML 2019.
>
> [2] Prasad, A., Suggala, A. S., Balakrishnan, S., & Ravikumar, P.. Robust Estimation via Robust Gradient Estimation. Journal of the Royal Statistical Society, Series B (Statistical Methodology) 2022.
>
> ---
>
> **R**: The paper does not include experimental results, which could limit the confidence in the practical effectiveness of the proposed approach.
>
> **A**: Our main contribution is to study the problem of phase retrieval in the $\epsilon$-corruption model and to provide a provably robust, nearly-sample-optimal, and nearly-linear time algorithm.
>
> Many influential ML conference papers had a large impact despite having no experiments. Moreover, provable theoretical guarantees are particularly important for designing robust machine learning algorithms. We believe our results are substantial even without experiments, and we hope that our theoretical results are judged based on their merits.
>
> ---
>
> **R**: The authors solely consider the scenario of noiseless intensity measurements and fail to take into account the noisy case.
>
> **A**: We chose to focus on the noiseless case to present our results and key ideas clearly, without any additional technical complexity. The current level of technical details in the paper is already substantial.
>
> We agree that extending our work to include broader families of distributions, allowing noise in measurements, and conducting experiments is an important and exciting avenue for future research.

---

> > ### Comment · Reviewer_BQYS · 2024-08-12
> > **Response to rebuttals**
> >
> > Thank you for the responses. I persist in considering that the theoretical results are somewhat incremental, yet they do not appear significant enough to warrant a purely theoretical submission. Experimental validations of the proposed algorithm (e.g., numerically verifying the counter-intuitive result that there is no dependence on $\epsilon$ in both the sample complexity and the upper bound ) would be advantageous for this submission. Therefore, I am inclined to maintain my current score.

---

### Official Review · Reviewer_XpSZ · 2024-07-12

**Soundness:** 3
**Presentation:** 4
**Contribution:** 3
**Rating:** 7
**Confidence:** 4

**Summary:**

This paper studies a classical problem called phase retrieval. The goal is to obtain unknown $d$-dimensional vector $x$ from $n$ datapoints $(a_i, \langle a_i, x \rangle^2)$. This work assumes that $a_i$ are iid Gaussian vectors, but also that a small $\varepsilon$ fraction of the data is corrupted. The authors suggest a two-stage process to identify the vector up to a small error. First, a spectral based algorithm is used to have a small constant error. Further, a robust gradient descent is used to approximate the initial guess up to a small error.

**Strengths:**

1. Paper is well-written and provides a good overview of the problem and of the techniques.
2. Phase retrieval is a traditional non-convex problem, which was largely studied before, and understanding how robust algorithms perform on it is important.
3. Paper uses prior technique in a simple way, and it is possible that this two-stage approach can be applied to other problems.

**Weaknesses:**

1. Results are limited to the Gaussian setting.
2. The method for RME that is used assumes that variance $\sigma$ is known? But in the way it is used here, $\sigma$ depends on the distance between current solution and the true vector. Authors do not comment on this issue.
3. $\tilde O, \tilde \Omega$ notation is not defined.
4.  Intuition in line 98 in my interpretation contradicts more exact version in line 214 (In the end, if I understand correctly, the crucial reason why the spectral initialization algorithm works is that the adversary cannot change the top eigendirection, but can only add new directions).

**Questions:**

1. How applicable are the techniques beyond Gaussian iid setting?
2. Why using $k = 1$ is not sufficient for the optimization problem in Algorithm 1?

**Limitations:**

There are no ethical limitations of this work.

---

> ### Author Rebuttal · Authors · 2024-08-06
>
> We appreciate the reviewer's careful review and feedback.
>
> ---
>
> **R**: The method for RME that is used assumes that variance $\sigma$ is known? But in the way it is used here, it depends on the distance between the current solution and the true vector. The authors do not comment on this issue.
>
> **A**: Thank you for pointing this out. We agree that this requires clarification and we will address this.
>
> An upper bound on the distance between the current solution $z$ and the ground-truth vector $x$ (i.e., an upper bound on $\sigma$) suffices for our proof. We can indeed maintain such an upper bound, which starts at $1/8$ after spectral initialization and decreases geometrically as proved in Lemma 5.3.
>
> ---
>
> **R**: How applicable are the techniques beyond the Gaussian iid setting?
>
> **A**: Our results can be extended to any distribution of sensing vectors $a_i$ that satisfy Lemma 4.2 and Lemma 5.1. This includes, for example, subgaussian distributions.
>
> Intuitively, Lemma 4.2 states that certain 4th-moment quantities $\sum_i y_i a_i a_i^\top$ do not change much after removing any $\epsilon$ fraction of the $a_i$’s. We will add discussions on this, thank you.
>
> ---
>
> **R**: Why using $k=1$ is not sufficient for the optimization problem in Algorithm 1?
>
> **A**: To match the desired spectrum $(3, 1, \ldots)$, where the largest eigenvalue is $3$ and all other eigenvalues are $1$, we need to minimize the sum of the first *two* eigenvalues.
>
> If we only minimize the largest eigenvalue, we could get a spectrum that looks like $(3, 3, 1, …)$ with no unique top eigenvector, which can be problematic. We will add discussions on this.
>
> ---
>
> **R**: $\tilde O$, $\tilde{\Omega}$ notation is not defined.
>
> **A**: Thank you for pointing this out. There was a footnote in previous versions that defined $\tilde O$ and $\tilde \Omega$, but it was deleted by accident. We will fix this.
>
> ---
>
> **R**: Discrepancy between intuition of line 98 and exact version in line 214.
>
> **A**: These statements do not contradict each other. Let $Y = \frac{1}{n} \sum_i y_i a_i a_i^\top$. Without corruption, the expectation of $Y$ is $I + 2 x x^\top$.
>
> The adversary can add arbitrary directions to $Y$. For example, he can change $Y$ to $I + 2 x x^\top + 100 b b^\top$ for some arbitrary unit vector $b$ that is orthogonal to $x$. The top eigenvector of $Y$ then becomes $b$.
>
> The key observation is that the adversary cannot erase the $2 x x^\top$ term due to stability conditions in Lemma 4.2. Therefore, by minimizing the sum of the top two eigenvalues, our algorithm can find weights such that the weighted sum $\sum_i w_i y_i a_i a_i^\top$ is close to $I + 2 x x^\top$.
>
> We will make this more clear, thank you for this feedback.

---

> > ### Comment · Reviewer_XpSZ · 2024-08-12
> >
> > Thank you for your detailed reply. I carefully read other reviews together with the authors' responses and remain in favour of acceptance for this work.
> > In contrast to other reviewers, I do not find the fact that the error / sample complexity does not depend on $\varepsilon'$ surprising: note that in robust mean estimation, even when $\varepsilon = 0$ (no adversarial data), it is impossible to recover mean *exactly*, whereas in the phase retrieval it is possible, as the authors responded in one of the comments.
> > Therefore, it is not counterintuitive that the algorithm proposed by the authors obtains accurate estimates independent from the corruption parameter.

---

### Official Review · Reviewer_MNuM · 2024-07-16

**Soundness:** 2
**Presentation:** 2
**Contribution:** 2
**Rating:** 3
**Confidence:** 4

**Summary:**

The authors study the phase retrieval problem for retrieving a real signal under the influence of arbitrary corruption. The corruption is allowed to be present in labels or features. They propose a two-step solution. First, they ensure that the initialization is robust to the corruption and second, they show that the gradient descent updates can be made resilient. Authors claim that their method can recover the true signal (with possibly sign mismatch) to an arbitrary precision.

**Strengths:**

The ideas presented in the paper are certainly interesting. If resilience to corruption can be achieved in individual steps, then it makes sense that it might lead to a good overall recovery.

**Weaknesses:**

1. The authors claim that corruption level up to some universal constant $\epsilon'$ can be handled through their method. Although, to the best of my understanding, this quantity is not characterized in the main paper. What is the maximum value for $\epsilon'$?
2. There is no discussion on the dependency of $\epsilon'$ on $n$ or $d$.
3. The claim of signal recovery to an arbitrary precision puzzles me. It is known that for $\epsilon$-corrupted vectors, the robust mean estimation can only be done up to $\Omega(\sqrt{\epsilon})$ error. Despite that, the authors claim signal recovery (with possibly a flipped sign) to an arbitrary precision. Can the authors comment on how this is achieved?
4. The claim in line 213 says that $y_i$ is always greater than $0$. Why is that true when the adversary can corrupt $y_i$ arbitrarily? As far as I can tell, the algorithm does not discard negative $y_i$s.

**Questions:**

Please see above.

**Limitations:**

Please see above.

---

> ### Author Rebuttal · Authors · 2024-08-06
>
> We thank the reviewer for their close reading of our work and their feedback.
>
> ---
>
> **R**: The authors claim that corruption levels up to some universal constant $\epsilon’$ can be handled through their method. Although, to the best of my understanding, this quantity is not characterized in the main paper. What is the maximum value for $\epsilon’$?
> There is no discussion on the dependency of $\epsilon’$ on $n$ or $d$.
>
> **A**: The universal constant $\epsilon’$ does not depend on $n$ and $d$. A “universal constant” is an absolute constant that does not depend on other parameters. We will clarify this.
>
> The assumption that the corruption level $\epsilon < \epsilon’$ is standard in the robust statistics literature, see e.g. [1][2][3], and the literature typically does not optimize the constant $\epsilon’$ in the proofs.
>
> While the maximum value for $\epsilon’$ needed for the theoretical analysis is usually quite small, e.g., $10^{-3}$, previous work [4] showed that these algorithms can tolerate up to $15\%$ corruption in some practical applications.
>
> [1] Diakonikolas, I., Kane, D. M., Pensia, A., & Pittas, T.. Streaming Algorithms for High-Dimensional Robust Statistics. ICML 2022.
>
> [2] Cheng, Y., Diakonikolas, I., Ge, R., & Woodruff, D. P.. Faster Algorithms for High-Dimensional Robust Covariance Estimation. COLT 2019.
>
> [3] Diakonikolas, I., & Kane, D. M.. Algorithmic High-Dimensional Robust Statistics (book). Cambridge University Press 2023.
>
> [4] Cheng, Y., Diakonikolas, I., Kane, D., & Stewart, A.. Robust Learning of Fixed-Structure Bayesian Networks. NeurIPS 2018.
>
> ---
>
> **R**: The claim of signal recovery to an arbitrary precision puzzles me. It is known that for
> $\epsilon$-corrupted vectors, the robust mean estimation can only be done up to
>  $\Omega(\sqrt{\epsilon})$ error. Despite that, the authors claim signal recovery (with possibly a flipped sign) to an arbitrary precision. Can the authors comment on how this is achieved?
>
> **A**: This is achievable because as our current solution $z$ gets closer to the true vector $x$, the covariance matrix of the true gradients $\Sigma_z$ becomes smaller. This is formally stated in our Lemma 5.1: $\Sigma_z \preceq O(\lVert x-z \rVert_2^2) I$.
>
> For robust mean estimation, if the clean samples are drawn from a distribution with unknown mean and unknown covariance matrix $\Sigma \preceq \sigma^2 I$, we can achieve an estimation error of $O(\sqrt{\epsilon} \sigma)$. As we get closer to the ground truth, $\sigma$ becomes smaller, which allows us to obtain increasingly precise gradient estimates, which enables recovery to arbitrary precision.
>
> We will add a discussion to clarify this in Section 1.2.
>
> ---
>
> **R**: The claim in line 213 says that $y_i$ is always greater than 0. Why is that true when the adversary can corrupt arbitrarily? As far as I can tell, the algorithm does not discard negative $y_i$s.
>
> **A**: Because $y_i = \langle a_i, x \rangle^2$, we know that any $y_i < 0$ must be corrupted. Thus, we can discard these input pairs $(a_i, y_i)$ and can assume without loss of generality that $y_i \ge 0$. Thank you for pointing this out and we will clarify this.

---

> > ### Comment · Reviewer_MNuM · 2024-08-10
> >
> > Thank you for your response. If you go through your suggested references, say [1, 2, 4], you will observe two important features in their theorem statements:
> > 1. Their sample complexities depend on the amount of corruption $\epsilon$.
> > 2. Their convergence results depend on the amount of corruption.
> > For example, check Theorem 1.3.1 or D.2 of [1], where $n = \mathcal{O}(\frac{d^2}{\epsilon})$ and $\\| \hat{\mu} - \mu_D \\| \leq \sqrt{\epsilon}$. This shows a clear dependency of sample complexity and the convergence rate on the amount of corruption. Your results claim recovery up to arbitrary precision with an unknown $\epsilon$. To justify such a strong statement, in my opinion, it is important to characterize $\epsilon'$ in terms of $n$ and $d$.

---

> ### Author Response · Authors · 2024-08-10
>
> In our work, we study the problem of robust phase retrieval, which is different from robust mean estimation.
>
> At a high level, robust phase retrieval has an information-theoretic lower bound of error $0$ (i.e., exact recovery), whereas robust mean estimation has an information-theoretic lower bound of $O(\sqrt{\epsilon})$.
>
> For robust phase retrieval, $O(d)$ samples are sufficient for exact recovery, even when $\epsilon$-fraction of the input is corrupted. This is possible due to the special structure of the input: $y_i = \langle a_i, x \rangle^2$. (Our contribution is to show that this can be done in nearly-linear time.)
>
> For robust mean estimation (for bounded covariance distributions in $\mathbb{R}^d$), as the reviewer correctly noted, the information-theoretic optimal error is $O(\sqrt{\epsilon})$, and it takes $O(d/\epsilon)$ samples to achieve this error.

---

> > ### Comment · Reviewer_MNuM · 2024-08-10
> >
> > Thank you for the response. Could you please provide the reference containing an information-theoretic lower bound for the phase-retrieval problem under corruption?

---

> ### Author Response · Authors · 2024-08-10
>
> Here is a proof that information-theoretically, $O(d)$ samples are sufficient for exact recovery for phase retrieval, even under $\epsilon$-corruption.
>
> -----
>
> Let $G^*$ denote the original $n$ clean input pairs $(a_i, y_i)$. Each pair specifies a constraint $y_i = \langle a_i, x\rangle^2$.
>
> We can assume that for any subset $S \subseteq G^*$ with $|S| \ge (1-2\epsilon)n$, only the ground-truth vector $\pm x$ can satisfy all constraints in $S$. Intuitively, this holds because $x$ has $d$ degrees of freedom, and we have $(1-2\epsilon)n > d$ constraints with essentially i.i.d. Gaussian $a_i$'s.
>
> Consider the following exponential-time algorithm: Given an $\epsilon$-corrupted set of $(a_i, y_i)$ pairs,
> * Enumerate all subset $T$ of size $(1-\epsilon)n$.
> * For each $T$, run a standard phase retrieval algorithm on $T$,
>   * If a solution $z$ is returned and $z$ satisfies all constraints in $T$, return $z$.
>
> First observe that this algorithm must return a solution, because $T$ will eventually hit the set of $(1-\epsilon)n$ clean pairs.
>
> Next observe that this algorithm cannot return anything other than $\pm x$, this is because $|T| = (1-\epsilon)n$ and there are at most $\epsilon n$ corrupted pairs, so $T$ contains at least $(1-2\epsilon)n$ clean pairs. As we assumed earlier, only $\pm x$ satisfies any subset of $(1-2\epsilon)n$ clean pairs.
>
> -----
>
> To our knowledge, there is no existing reference for this specific information-theoretic lower bound. (Although the above exponential-time algorithm is forklore in robust statistics.) We are the first to study this problem. Our paper provides another proof that exact recovery is possible with $O(d)$ samples (and we show how to achieve it in nearly-linear time).

---

> > ### Comment · Reviewer_MNuM · 2024-08-13
> >
> > Thank you for your response. However, I still have reservations about exact recovery in the constant corruption proportion case with a strong adversary model. Intuitively, a strong adversary could make an $\epsilon$ proportion of inputs identical for inputs coming from two different distributions. These distributions would be, at most, $1 - \epsilon$ away from each other in terms of total variation distance. This is problematic when $\epsilon$ is a constant (non-vanishing). Let me elaborate.
> >
> > Consider $\epsilon \in (0, 1]$ to be the constant corruption proportion. We will use the following phase retrieval model in one dimension:
> > $$ y = (x \theta)^2 + w_{\theta, x}$$
> > where $w_{\theta, x}$ denotes the adversarial corruption added by a strong adversary who has access to both $x$ and $\theta$. We draw $x$ from a standard normal distribution. Consider two parameters $\theta_1 > 0$ and $\theta_2 > 0$ with $| \theta_1 - \theta_2 | > \delta$ for some $\delta > 0$. Let $D_1$ and $D_2$ be distributions over $\mathbb{R} \times \mathbb{R}$ corresponding to linear models $y = (x\theta_1)^2 + w_{\theta_1, x}$ and $y = (x\theta_2)^2 + w_{\theta_2, x}$ respectively. Since the adversary can only change $\epsilon$ fraction of inputs, we assume the following marginal distribution for y for $i \in \\{1, 2\\}$:
> > $$D_i(y | x) = \begin{cases} 1 - \epsilon, \text{ when } y = (x\theta_i)^2 \\\\ \frac{\epsilon}{\sigma}, \text{ when } y \in [ \sigma,  2\sigma] \\\\ 0, \text{ otherwise} \end{cases}$$
> >
> > We want to be able to differentiate between $D_1$ and $D_2$ based on the inputs drawn from either $D_1$ or $D_2$. By reduction to a hypothesis testing problem and using Neyman-Pearson lemma:
> > $$ \inf_{\hat{\theta}} \sup_{\theta \in \Theta} P[|\hat{\theta} - \theta|^2 > \delta^2] \geq \frac{1}{2}(1 - TV(D_1, D_2))$$
> > where $TV(D_1, D_2)$ is the total variation difference between distributions $D_1$ and $D_2$.
> >
> > In our illustrative example,
> > $$TV(D_1, D_2) = \frac{1}{2}\int_{\mathbb{R} \times \mathbb{R}} |D_1(x, y) - D_2(x, y)|$$
> > $$TV(D_1, D_2) = \frac{1}{2}\int_{\mathbb{R} \times \mathbb{R}} D_1(x) |D_1(y|x) - D_2(y|x)|$$
> >
> > Notice that $D_1(y|x)$ and $D_2(y|x)$ can only differ when $(x\theta_1)^2 \ne (x\theta_2)^2$ and contribute $|D_1(y|x) - D_2(y|x)| \leq 2(1 - \epsilon)$ correspondingly. Overall,
> >
> > $$TV(D_1, D_2) \leq 1 - \epsilon$$
> >
> > It follows that,
> > $$ \inf_{\hat{\theta}} \sup_{\theta \in \Theta} P[|\hat{\theta} - \theta|^2 > \delta^2] \geq \frac{\epsilon}{2}$$
> >
> > Note that $\epsilon$ is constant, and this minimax rate seems unavoidable to me. Am I missing something here?

---

> > > ### Author Response · Authors · 2024-08-13
> > >
> > > Thank you for your detailed follow-up. We will respond using the setting and notations you provided.
> > >
> > > In the 1-D case, let $P_1$ and $P_2$ denote the clean distributions corresponding to the ground-truth parameters $\theta_1$ and $\theta_2$ respectively. Specifically, $(x, y) \sim P_1$ is distributed as first drawing $x \sim \mathcal{N}(0, 1)$, and then setting $y = (\theta_1 x)^2$. The distribution $P_2$ is defined similarly where $y = (\theta_2 x)^2$.
> > >
> > > As long as $\theta_1 \neq \pm \theta_2$, the total variation distance between $P_1$ and $P_2$ is always $1$. These two distributions have essentially disjoint support. This means that an adversary cannot make $P_1$ and $P_2$ indistinguishable by corrupting an $\epsilon$-fraction of the samples.
> > >
> > > In particular, if the corruption level is less than half, one can compute the (multi-)set $S = \{\sqrt{y_i/x_i^2}\}$, and the most frequent value in $S$ will be the correct $\theta$ with probability $1$.

---

> > > > ### Author Response · Authors · 2024-08-14
> > > >
> > > > The TV-distance between $P_1$ and $P_2$ is always $1$ as long as $\theta_1 \neq \theta_2$, as a result the TV-distance between $D_1$ and $D_2$ cannot be less than $1-2\epsilon$ (you gave an upper bound of $1-\epsilon$ but what is more important is the lower bound), so in the 1-D case, we can achieve exact recovery with for example $n \ge 3$ samples when $\epsilon \le 1/3$.
> > > >
> > > > Note that our task is not to distinguish whether a single sample come from $D_1$ or $D_2$. Our task (the property testing version) is, given an $\epsilon$-corrupted set of $O(d)$ samples drawn from either $D_1$ or $D_2$, decide whether they are drawn from $D_1$ or $D_2$ with high constant probability (which is what we claim in Theorem 3.1).

---

### Decision · Program_Chairs · 2024-09-25

**Decision:**

Reject

**Comment:**

The paper studies a variant of phase retrieval, in which we observe y_i = |<a_i, x >| and the goal is to recover x. Here, a fraction of the pairs (a_i, y_i) can be adversarially corrupted. The paper proposes a two-phase algorithm which deploys tools of robust statistics in each phase. In the first, initialization, phase, the proposal is to modify the standard spectral initialization by weights which control the influence of adversarially modified pairs (a_i,y_i) on the lead eigenvector. These weights are chosen by solving a certain SDP, which admits efficient special-purpose solvers. In the second, estimation phase, the paper deploys a robust mean estimator to reduce the effect of outlying samples on the gradient.

Reviewers produced a mixed evaluation of the paper. On the positive side, the paper analyzes robust phase retrieval under a novel setting, in which both a_i and y_i can be modified; while a number of past works utilize robust estimators for nonconvex robust phase retrieval, these works focus on corrupted measurements y_i, rather than pairs (a_i,y_i). The paper is crisply written. Reviewers raised questions about the role of the corruption level \epsilon, the absence of experimental validation and the relationship to the existing literature on phase retrieval. The authors convincingly addressed questions about the level of corruption; it is indeed a standard assumption in robust statistics that the corruption level is bounded by an absolution constant. Other reviewer concerns centered around the significance of this formulation of robust phase retrieval vis-a-vis existing models, which could be better addressed in future versions of this submission. After considering author feedback, reviewers retained a mixed evaluation of the work, putting it below the bar for acceptance.